# MATRIX INFORMATION THEORY FOR SELF-SUPERVISED LEARNING

## ABSTRACT

Contrastive learning often relies on comparing positive anchor samples with multiple negative samples to perform Self-Supervised Learning (SSL). However, non-contrastive approaches like BYOL, SimSiam, and Barlow Twins achieve SSL without explicit negative samples. In this paper, we introduce a unified matrix information-theoretic framework that explains many contrastive and non-contrastive learning methods. We then propose a novel method Matrix-SSL based on matrix information theory. Experimental results reveal that Matrix-SSL significantly outperforms state-of-the-art methods on the ImageNet dataset under linear evaluation settings and on MS-COCO for transfer learning tasks. Specifically, when performing 100 epochs pre-training, our method outperforms SimCLR by 4.6%, and when performing transfer learning tasks on MS-COCO, our method outperforms previous SOTA methods such as MoCo v2 and BYOL up to 3.3% with only 400 epochs compared to 800 epochs pre-training.

## 1 INTRODUCTION

Self-supervised learning (SSL) has achieved remarkable performance on various tasks like image classifications and segmentations and even outperforms the supervised counterparts (Chen et al., 2020a; Caron et al., 2021; Li et al., 2021; Zbontar et al., 2021; Bardes et al., 2021). Contrastive learning is an important type of self-supervised learning method. For example, SimCLR (Chen et al., 2020a) uses InfoNCE (Oord et al., 2018) loss to do contrastive self-supervised learning, and has been proved to exactly perform spectral clustering on similarity graph (Tan et al., 2023). Spectral contrastive loss (HaoChen et al., 2021) is a variant of SimCLR and also has been proven to be doing spectral clustering under similar settings. The empirical success of contrastive learning has spurred a number of theoretical explorations into the contrastive loss (Arora et al., 2019; HaoChen et al., 2022; Tosh et al., 2020; 2021; Lee et al., 2020; Wang et al., 2022b; Nozawa & Sato, 2021; Huang et al., 2021; Tian, 2022; Hu et al., 2022).

Recently, many researchers have been exploring non-contrastive self-supervised learning without explicit negative samples using various different losses (BYOL (Grill et al., 2020), SimSiam (Chen & He, 2021), Barlow Twins (Zbontar et al., 2021), VICReg (Bardes et al., 2021), MEC (Liu et al., 2022), etc). Many theoretical studies have also paid attention to the intrinsic mechanism behind non-contrastive SSL methods. For example, Garrido et al. (2022) established the duality between contrastive and non-contrastive methods, and Balestriero & LeCun (2022) unveiled the links of some contrastive and non-contrastive methods to spectral methods.

In this paper, we propose a unified matrix information-theoretic framework to analyze contrastive and non-contrastive learning methods. Our investigation starts from the observation that loss functions of these methods are mostly based on $\mathbf{Z}_1$ and $\mathbf{Z}_2$, two feature representation matrices derived from data augmentations. Inspired by this, we extend classical concepts like entropy, KL divergence, and cross-entropy to their matrix counterparts, facilitating a more expressive representation of the associated loss functions.

In particular, from the matrix cross-entropy (MCE) perspective, SimCLR loss is optimizing the diagonal of MCE, while non-contrastive learning methods like SimSiam or BYOL either aim to maximize some trace norm or strive to align the covariance matrix closely with the identity matrix. Interestingly, previous methods like Barlow Twins or MEC can also be succinctly derived as loss functions based on MCE, a connection not obvious in their original descriptions.

Leveraging this novel framework, we propose Matrix-SSL, a natural extension of the uniformity and alignment idea from Wang & Isola (2020). This algorithm comprises two components of loss functions: Matrix-KL-Uniformity and Matrix-KL-Alignment. The former employs MCE to align the cross-covariance matrix between $\mathbf{Z}_1$ and $\mathbf{Z}_2$ with the identity matrix $\mathbf{I}_d$, while the latter directly aligns the auto-covariance matrix of $\mathbf{Z}_1$ and $\mathbf{Z}_2$.

In experimental evaluations, our method Matrix-SSL outperforms state-of-the-art methods (SimCLR, BYOL, SimSiam, Barlow Twins, VICReg, etc.) by a large margin on ImageNet datasets, especially under linear evaluation settings, our method uses only 100 epochs pre-training can outperform SimCLR 100 epochs pre-training by 4.6%. For transfer learning tasks such as COCO detection and COCO instance segmentation, our method outperforms previous SOTA methods such as MoCo v2 and BYOL up to 3% with only 400 epochs compared to 800 epochs pre-training.

In summary, our contributions can be listed as three-fold:

- We present a matrix information-theoretic interpretation of uniformity pursuit in Self-Supervised Learning.
- Rooted in matrix information theory, we propose matrix-alignment loss and Matrix-SSL to directly optimize matrix KL divergence and matrix cross-entropy, and outperform SOTA methods on linear probing and COCO transfer learning tasks.
- We examine the rank-increasing phenomenon, and link it to the matrix entropy and uniformity pursuit, then explain the effectiveness of our method.

## 2 RELATED WORK

**Contrastive and non-contrastive SSL approaches.** Contrastive and non-contrastive self-supervised learning methods learn representations based on diverse views or augmentations of inputs, and they do not rely on human-annotated labels (Chen et al., 2020a; Hjelm et al., 2018; Wu et al., 2018; Tian et al., 2019; Chen & He, 2021; Gao et al., 2021; Bachman et al., 2019; Oord et al., 2018; Ye et al., 2019; Henaff, 2020; Misra & Maaten, 2020; Caron et al., 2020; HaoChen et al., 2021; Caron et al., 2021; Li et al., 2021; Zbontar et al., 2021; Tsai et al., 2021b; Bardes et al., 2021; Tian et al., 2020; Robinson et al., 2021; Dubois et al., 2022; Kim et al., 2023). These representations can be used for various of downstream tasks, achieving remarkable performance and even outperforming supervised feature representations. In our work, we propose to directly minimize the matrix KL divergence and matrix cross-entropy in the pursuit of alignment and uniformity, with strong minimization property and convexity.

**Theoretical understanding of self-supervised learning.** The empirical success of contrastive learning has triggered a surge of theoretical explorations into the contrastive loss (Arora et al., 2019; HaoChen et al., 2021; 2022; Tosh et al., 2020; 2021; Lee et al., 2020; Wang et al., 2022b; Nozawa & Sato, 2021; Huang et al., 2021; Tian, 2022; Hu et al., 2022; Tan et al., 2023). Wang & Isola (2020) shed light on the optimal solutions of the InfoNCE loss, decomposing it as alignment term and uniformity term, contributing to a deeper comprehension of self-supervised learning. In HaoChen et al. (2021); Shen et al. (2022); Wang et al. (2022b); Tan et al. (2023), self-supervised learning methods are examined from a spectral graph perspective. Alongside these black-box interpretations of contrastive learning, Saunshi et al. (2022); HaoChen & Ma (2022) argue that inductive bias also has significant influences on the downstream performance of self-supervised learning. Cabannes et al. (2023) introduces a theoretical framework that elucidates the intricate relationship among the selection of data augmentation, the design of network architecture, and the training algorithm.

Various theoretical studies have also investigated non-contrastive methods for self-supervised learning (Wen & Li, 2022; Tian et al., 2021; Garrido et al., 2022; Balestriero & LeCun, 2022; Tsai et al., 2021b; Pokle et al., 2022; Tao et al., 2022; Lee et al., 2021). Garrido et al. (2022) establishes the duality between contrastive and non-contrastive methods. Balestriero & LeCun (2022) reveal the connections between variants of SimCLR, Barlow Twins, and VICReg to ISOMAP, Canonical Correlation Analysis, and Laplacian Eigenmaps, respectively. The fact that a method like SimSiam does not collapse is studied in Tian et al. (2021). The loss landscape of SimSiam is also compared to SimCLR's in Pokle et al. (2022), which shows that it learns bad minima. A variant of Barlow Twins' criterion is also linked to a variant of HSIC in Tsai et al. (2021b). The use of data augmentation in sample-contrastive learning has also been studied from a theoretical standpoint in Huang et al. (2021);

Wen & Li (2021). In our work, we present a unified matrix information-theoretic understanding of several renowned self-supervised learning methods from the matrix-information-theoretic uniformity and alignment pursuit.

**Neural collapse and dimensional collapse.** Papyan et al. (2020) describe the intriguing phenomenon of Neural Collapse (NC), which manifests when training a classification network with cross-entropy loss. This phenomenon can be summarized that all the features of a single class converge to the mean of these features. In addition, the class-means form a simplex equiangular tight frame (ETF). Zhuo et al. (2023) advocate for a comprehensive theoretical understanding of non-contrastive learning through the mechanism of rank differential. We briefly introduce NC and ETF in Appendix D. In our work, we discovered the exact closed-form relationships among effective rank, matrix entropy, and other matrix information-theoretic quantities.

## 3 BACKGROUND

Self-supervised learning (SSL) aims to learn meaningful representations from unlabeled data $\{x_i\}_{i=1}^n$, which can be used to enhance performance in various downstream tasks. Prominent SSL architectures like SimCLR, SimSiam, BYOL, Barlow Twins, VICReg, etc. employ 2-view (can be generalized into multi-view) augmentations: an online network $\boldsymbol{f}_\theta$ and a target network $\boldsymbol{f}_\phi$. Given a mini-batch $\{\mathbf{x}_i\}_{i=1}^B$, each data point $\mathbf{x}_i$ is augmented using a random transformation $\mathcal{T}$ from a predefined set $\tau$ to obtain $\mathbf{x}_i' = \mathcal{T}(\mathbf{x}_i)$. These original and augmented data points are processed through the respective networks to generate feature representations $\mathbf{z}_1^i$ and $\mathbf{z}_2^i$, both residing in $\mathbb{R}^d$. The resulting matrices $\mathbf{Z}_1$ and $\mathbf{Z}_2 \in \mathbb{R}^{d \times B}$ form the basis for the training loss $\mathcal{L}(\mathbf{Z}_1, \mathbf{Z}_2)$, which varies based on the learning paradigm—contrastive or non-contrastive. In this work, we extend this framework through matrix information theory, providing a unified understanding of both paradigms.

### 3.1 CONTRASTIVE SSL APPROACHES

The idea of contrastive learning is to make the representation of similar objects align and dissimilar objects apart. One of the widely adopted losses in contrastive learning is the SimCLR (Chen et al., 2020a) (InfoNCE (Oord et al., 2018)) loss which is defined as follows (where we use cosine similarity $\text{sim}(\boldsymbol{u}, \boldsymbol{v}) = \boldsymbol{u}^\top \boldsymbol{v} / (\|\boldsymbol{u}\|_2 \|\boldsymbol{v}\|_2)$):

$$\mathcal{L}_{\text{SimCLR}}(\mathbf{z}_i, \mathbf{z}_j) = -\log \frac{\exp\left(\text{sim}\left(\boldsymbol{z}_i, \boldsymbol{z}_j\right)/\tau\right)}{\sum_{k=1}^{2B} \mathbf{1}_{[k \neq i]} \exp\left(\text{sim}\left(\boldsymbol{z}_i, \boldsymbol{z}_k\right)/\tau\right)}. \tag{1}$$

Recently, Tan et al. (2023) have proved that SimCLR is essentially running spectral clustering on similarity graph (induced by augmentation process), while HaoChen et al. (2021) have proved that spectral contrastive loss is also performing spectral clustering on augmentation graph.

### 3.2 NON-CONTRASTIVE SSL APPROACHES

SimSiam (Chen & He, 2021) employ negative cosine similarity as its loss function, which can also be considered as the mean squared error of $\ell_2$-normalized vectors $\mathbf{z}_v^i$ ($i \in \{1, \cdots, B\}, v \in \{1, 2\}$) which are used in BYOL (Grill et al., 2020):

$$\mathcal{L}_{\text{SimSiam}} = -\sum_{i=1}^B {\mathbf{z}_1^i}^\top \mathbf{z}_2^i = -\text{tr}\left(\mathbf{Z}_1^\top \mathbf{Z}_2\right) = -\text{tr}\left(\mathbf{Z}_1 \mathbf{Z}_2^\top\right). \tag{2}$$

$$\mathcal{L}_{\text{BYOL}} = \sum_{i=1}^B \|\mathbf{z}_1^i - \mathbf{z}_2^i\|_2^2 = 2B - 2\text{tr}\left(\mathbf{Z}_1^\top \mathbf{Z}_2\right) = 2B - 2\text{tr}\left(\mathbf{Z}_1 \mathbf{Z}_2^\top\right). \tag{3}$$

Barlow Twins (Zbontar et al., 2021) and VICReg (Bardes et al., 2021) aim for similar objectives: making the cross-correlation matrix as close to the identity matrix as possible, while also reducing redundancy among features. The Barlow Twins loss is:

$$\mathcal{L}_{\mathcal{BT}} = \sum_{i=1}^d \left(1 - \left(\mathbf{Z}_1 \mathbf{Z}_2^\top\right)_{ii}\right)^2 + \lambda_{\mathcal{BT}} \sum_{i=1}^d \sum_{j \neq i}^d \left(\mathbf{Z}_1 \mathbf{Z}_2^\top\right)_{ij}^2. \tag{4}$$

The loss function of Variance-Invariance-Covariance Regularization (VICReg) by Bardes et al. (2021) can be formalized as:

$$\mathcal{L}_{\text{VICReg}} = \alpha \sum_{k=1}^{d} \max\left(0, 1 - \sqrt{\text{Cov}(\mathbf{Z}, \mathbf{Z})_{k,k}}\right) + \beta \sum_{j=1}^{d} \sum_{k \neq j}^{d} \text{Cov}(\mathbf{Z}, \mathbf{Z})_{k,j}^2$$
$$+ \frac{\gamma}{N} \sum_{i=1}^{B} \sum_{j=1}^{B} (\boldsymbol{G})_{i,j} \left\| \mathbf{Z}_{i,.} - \mathbf{Z}_{j,.} \right\|_2^2. \tag{5}$$

Where Cov denotes the covariance matrix. Here we use the notations from Balestriero & LeCun (2022), denoting a known pairwise positive relation between samples in the form of a symmetric matrix $\boldsymbol{G} \in \{0,1\}^{N \times N}$ where $(\boldsymbol{G})_{i,j} = 1$ iff samples $\boldsymbol{x}_i$ and $\boldsymbol{x}_j$ are known to be semantically related, and with 0 in the diagonal.

The Total Coding Rate (TCR) and MCR$^2$ method (Ma et al., 2007; Li et al., 2022; Tong et al., 2023), which is grounded in coding theory and compression theory, has following loss function:

$$\mathcal{L}_{\text{TCR}} = -\frac{1}{2} \log \det \left( \mathbf{I}_d + \frac{d}{B\epsilon^2} \mathbf{Z}\mathbf{Z}^\top \right), \tag{6}$$

Recent work ColorInfomax (Ozsoy et al., 2022) aims at capturing the linear dependence among alternative representations, their loss function can be written down as:

$$\mathcal{L}_{\text{ColorInfoMax}} = -\log \det \left( \text{Cov}(\mathbf{Z}_1, \mathbf{Z}_1) + \varepsilon\mathbf{I} \right) - \log \det \left( \text{Cov}(\mathbf{Z}_2, \mathbf{Z}_2) + \varepsilon\mathbf{I} \right)$$
$$- \frac{2}{\varepsilon B} \left\| \mathbf{Z}^{(1)} - \mathbf{Z}^{(2)} \right\|_F^2. \tag{7}$$

Given that both the online and target networks are approximations of the feature map $\boldsymbol{f}$, we can use the cross-covariance between $\mathbf{Z}_1$ and $\mathbf{Z}_2$ to approximate $\mathbf{Z}\mathbf{Z}^\top$, resulting in the maximal entropy coding (MEC) loss (Liu et al., 2022):

$$\mathcal{L}_{\text{MEC}} = -\mu \log \det \left( \mathbf{I}_d + \frac{d}{B\epsilon^2} \mathbf{Z}_1\mathbf{Z}_2^\top \right) = -\mu \, \text{tr} \left( \log \left( \mathbf{I}_d + \frac{d}{B\epsilon^2} \mathbf{Z}_1\mathbf{Z}_2^\top \right) \right), \tag{8}$$

Another possible formulation for the multi-views (Siamese as 2-views) network architecture could be to use the concatenated representations of different views (Tong et al., 2023):

$$\mathcal{L}_{\text{EMP-TCR}} = -\mu \log \det \left( \mathbf{I}_d + \frac{d}{B\epsilon^2} \tilde{\mathbf{Z}}\tilde{\mathbf{Z}}^\top \right). \tag{9}$$

where $\tilde{\mathbf{z}}^i = [\mathbf{z}_1^{i^\top}, \mathbf{z}_2^{i^\top}, \cdots, \mathbf{z}_n^{i^\top}]^\top \in \mathbb{R}^{nd}$ is the concatenated representation (row vector) of $n$ views for image $\mathbf{x}_i$, and $\tilde{\mathbf{Z}} = [\tilde{\mathbf{z}}^1, \tilde{\mathbf{z}}^2, \cdots, \tilde{\mathbf{z}}^B]$.

### 3.3 Matrix Information-Theoretic Quantities

**Definition 3.1** (Matrix entropy for positive semi-definite matrices). For a positive semi-definite matrix $\mathbf{A} \in \mathbb{R}^{n \times n}$, the matrix entropy is defined as:

$$\text{ME}(\mathbf{A}) = -\text{tr}(\mathbf{A} \log \mathbf{A}) + \text{tr}(\mathbf{A}) = -\sum_{i=1}^{n} \lambda_i \log \lambda_i + \sum_i \lambda_i.$$

where $\log$ denotes the principal matrix logarithm (Higham, 2008). For zero eigenvalues, we define $\log(0) := 0$. Our proposed matrix entropy generalizes the definition of von Neumann entropy (von Neumann, 1932; Witten, 2020), which is restricted to density matrices with unit trace: $\text{VNE}(\hat{\mathbf{A}}) = -\text{tr}(\hat{\mathbf{A}} \log \hat{\mathbf{A}})$, s.t. $\text{tr}(\hat{\mathbf{A}}) = 1$.

**Definition 3.2** (Matrix KL divergence for positive semi-definite matrices). For two positive semi-definite matrices $\mathbf{P}, \mathbf{Q} \in \mathbb{R}^{n \times n}$, the matrix KL divergence is defined as:

$$\text{MKL}(\mathbf{P}||\mathbf{Q}) = \text{tr}(\mathbf{P} \log \mathbf{P} - \mathbf{P} \log \mathbf{Q} - \mathbf{P} + \mathbf{Q}). \tag{10}$$

Our matrix KL divergence generalizes the definition of quantum (von Neumann) KL divergence (relative entropy) (von Neumann, 1932; Witten, 2020; Bach, 2022), which is also restricted to density matrix with unit trace: $\mathrm{QKL}(\hat{\mathbf{A}}||\hat{\mathbf{B}}) = -\mathrm{tr}\left(\hat{\mathbf{A}}\log\hat{\mathbf{A}} - \hat{\mathbf{A}}\log\hat{\mathbf{B}}\right)$, s.t. $\mathrm{tr}(\hat{\mathbf{A}}) = \mathrm{tr}(\hat{\mathbf{B}}) = 1$.

In the same spirit that classical cross-entropy is often easier to optimize compared to classical KL divergence, we introduce the matrix cross-entropy:

**Definition 3.3** (Matrix Cross-Entropy (MCE) for positive semi-definite matrices). For two positive semi-definite matrices $\mathbf{P}, \mathbf{Q} \in \mathbb{R}^{n\times n}$, the matrix cross-entropy is defined as:

$$\mathrm{MCE}(\mathbf{P}, \mathbf{Q}) = \mathrm{MKL}(\mathbf{P}||\mathbf{Q}) + \mathrm{ME}(\mathbf{P}) = \mathrm{tr}(-\mathbf{P}\log\mathbf{Q} + \mathbf{Q}). \tag{11}$$

**Lemma 3.4.** *For any non-zero matrix* $\mathbf{A} \in \mathbb{R}^{m\times n}$, $\mathbf{A}\mathbf{A}^\top$ *is positive semi-definite.*

*Proof.* If not specified, we present proofs in the Appendix C. $\qquad\square$

Notice that we generalize quantum information quantities into positive semi-definite matrices without unit trace constraints, and the trio of matrix entropy, matrix KL divergence, and matrix cross-entropy exactly mirror the classical Shannon information-theoretic quantities, respectively.

### 3.4 Effective Rank

Roy & Vetterli (2007) introduced the concept of effective rank, which provides a real-valued extension of the classical rank.

**Definition 3.5** (Effective rank (Roy & Vetterli, 2007)). The effective rank of a non-all-zero $\mathbf{A} \in \mathbb{C}^{n\times n}$, denoted $\mathrm{erank}(\mathbf{A})$, is defined as

$$\mathrm{erank}(\mathbf{A}) \triangleq \exp\left\{\mathrm{H}\left(p_1, p_2, \ldots, p_n\right)\right\}, \tag{12}$$

where $p_i = \frac{\sigma_i}{\sum_{k=1}^n \sigma_k}$, $\{\sigma_i | i = 1, \cdots, n\}$ are the singular values of $\mathbf{A}$, and $\mathrm{H}$ is the Shannon entropy defined as $\mathrm{H}(p_1, p_2, \ldots, p_n) = -\sum_{i=1}^n p_i \log p_i$, with the convention that $0\log 0 \triangleq 0$.

We now then provide a proposition that closely relates effective rank and matrix information-theoretic quantities, suppose we have $l_2$ normalized representations $\mathbf{Z} = [\boldsymbol{f}(x_1), \cdots, \boldsymbol{f}(x_B)] \in \mathbb{R}^{d\times B}$.

**Proposition 3.6.** *Matrix entropy is a special case of matrix KL divergence to the uniform distribution* $\frac{1}{d}\mathbf{I}_d$, *and it has the following equality with connection to effective rank.*

$$\mathrm{ME}(\frac{1}{B}\mathbf{Z}\mathbf{Z}^\top) = -\mathrm{KL}(\frac{1}{B}\mathbf{Z}\mathbf{Z}^\top \,||\, \frac{1}{d}\mathbf{I}_d) + \log d + 1,$$

$$\mathrm{erank}(\frac{1}{B}\mathbf{Z}\mathbf{Z}^\top) = \exp\left(\mathrm{ME}(\frac{1}{B}\mathbf{Z}\mathbf{Z}^\top) - 1\right) = \exp\left(\mathrm{VNE}(\frac{1}{B}\mathbf{Z}\mathbf{Z}^\top)\right).$$

## 4 Matrix Information-Theoretic Perspectives of Self-Supervised Learning

Building upon the spectral clustering property inherent in contrastive learning—most notably in the SimCLR and spectral contrastive loss (HaoChen et al., 2021; Tan et al., 2023), which means that contrastive learning is more suitable for performing classification tasks other than dense feature tasks such as object detection and instance segmentation, which has been empirically validated by (Hua, 2021; Grill et al., 2020), we want to design other loss functions that can be used to get diverse features for dense prediction tasks.

We employ matrix KL divergence and matrix cross-entropy (MCE) as canonical metrics for comparing positive semi-definite matrices since they have strong minimization properties as shown below.

**Proposition 4.1** (Minimization property of matrix KL divergence).

$$\mathrm{argmin}_{\mathbf{Q}\succ 0}\,\mathrm{MKL}(\mathbf{P}\,||\,\mathbf{Q}) = \mathbf{P}. \tag{13}$$

**Proposition 4.2** (Minimization property of matrix cross-entropy)**.**

$$\mathrm{argmin}_{\mathbf{Q} \succ 0} \mathrm{MCE}(\mathbf{P}, \mathbf{Q}) = \mathbf{P}. \tag{14}$$

**Proposition 4.3** (Balestriero & LeCun (2022); Van Assel et al. (2022)). *SimCLR (InfoNCE) loss can be seen as matrix cross-entropy with an element-wise logarithm. When the relation matrix is a diagonal matrix, SimCLR's element-wise logarithm loss is equivalent to matrix cross-entropy (where* $\mathbf{G}$ *is defined in Section 3.2).*

$$\mathcal{L}_{SimCLR}^{diag} = -\sum_{i=1}^{2B} (\mathbf{G})_{i,i} \log\left(\widehat{\boldsymbol{G}}(\boldsymbol{Z})_{i,i}\right) = -\mathrm{tr}\left((\mathbf{G}^{diag}) \log\left(\widehat{\boldsymbol{G}}^{diag}(\boldsymbol{Z})\right)\right). \tag{15}$$

From Proposition 4.3, we find that SimCLR loss is not canonical for achieving matrix information-theoretic uniformity unless the relation matrix is diagonal (which is explicitly presented in the objective of de-correlation-based methods such as Barlow Twins and VICReg), we consider a natural way to achieve this without resorting to feature decorrelation and whitening methods, directly optimizing the covariance (auto-correlation) matrices.

Given a batch of $B$ data points $\{x_i\}_{i=1}^B$, and their $l_2$ normalized representations $\mathbf{Z} = [\boldsymbol{f}(x_1), \cdots, \boldsymbol{f}(x_B)] \in \mathbb{R}^{d \times B}$, from Lemma 3.4, we know that matrix information-theoretic quantities are well-defined. We design the following loss function to pursue uniformity, resulting in the following $\lambda$-regularized ($\lambda \geq 0$) Uniformity-MCE loss:

$$\mathcal{L}_{\mathrm{UMCE}} = \mathrm{MCE}\left(\frac{1}{d}\mathbf{I}_d + \lambda\mathbf{I}_d, \frac{1}{B}\mathbf{Z}\mathbf{Z}^\top + \lambda\mathbf{I}_d\right), \tag{16}$$

This MCE-based uniformity loss definition and its Matrix-KL divergence based counterpart are closely related are closely related to the Total Coding Rate (TCR) and MCR$^2$ method (Ma et al., 2007; Li et al., 2022; Tong et al., 2023), which is grounded in coding theory and compression theory. (Notice that in our experiments, the $\lambda$ in MCE loss is robust and can even be set as 0.)

**Theorem 4.4.** *Given a batch of $B$ data points $\{x_i\}_{i=1}^B$, and their $l_2$ normalized representations* $\mathbf{Z} = [\boldsymbol{f}(x_1), \cdots, \boldsymbol{f}(x_B)] \in \mathbb{R}^{d \times B}$, *define $\lambda$-regularized ($\lambda \geq 0$) Uniformity-MKL loss $\mathcal{L}_{UMKL}$ as:*

$$\mathcal{L}_{UMKL} = \mathrm{MKL}\left(\frac{1}{d}\mathbf{I}_d + \lambda\mathbf{I}_d \,\Big|\Big|\, \frac{1}{B}\mathbf{Z}\mathbf{Z}^\top + \lambda\mathbf{I}_d\right), \tag{17}$$

*Assume that $\lambda = \frac{\epsilon^2}{d}$ in TCR loss 6. Then, $\mathcal{L}_{UMCE}$ and $\mathcal{L}_{UMKL}$ can be expressed in terms of $\mathcal{L}_{TCR}$ as:*

$$\mathcal{L}_{UMKL} = (1 + d\lambda)(\log\frac{1+d\lambda}{\lambda d} + 2\mathcal{L}_{TCR}),$$
$$\mathcal{L}_{UMCE} = (1 + d\lambda)\left(-\log\lambda + 1 + 2\mathcal{L}_{TCR}\right).$$

This proposition encapsulates the relationship between Uniformity-MCE, Uniformity-MKL, and TCR losses, showing they are equivalent, up to constant terms and factors. From Proposition 4.1, 4.2, joint convexity of matrix KL (Lindblad, 1974) and Theorem 4.4, we have the following theorem.

**Theorem 4.5** (Minimization property of TCR loss)**.** *Given a batch of $B$ data points $\{x_i\}_{i=1}^B$, and their $l_2$-normalized representations $\mathbf{Z} = [\boldsymbol{f}(x_1), \cdots, \boldsymbol{f}(x_B)] \in \mathbb{R}^{d \times B}$, the global and unique minimizer under the constraint $\|\mathbf{z}_i\|_2^2 = 1$, for $i \in \{1, 2, \cdots, B\}$ of TCR loss is $\frac{1}{B}\mathbf{Z}\mathbf{Z}^\top = \frac{1}{d}\mathbf{I}_d$.*

**Proposition 4.6** (Taylor expansion of TCR loss (Liu et al., 2022))**.** *The Taylor series expansion of the Total Coding Rate (TCR) loss $\mathcal{L}_{TCR}$ around the zero matrix $\mathbf{0}$ can be expressed as:*

$$\mathcal{L}_{TCR}(\mathbf{Z}) = -\frac{d}{2B\epsilon^2}\mathrm{tr}(\mathbf{Z}\mathbf{Z}^\top) + \frac{d^2}{4B^2\epsilon^4}\mathrm{tr}(\mathbf{Z}\mathbf{Z}^\top\mathbf{Z}\mathbf{Z}^\top) + O\left(\left(\frac{d}{B\epsilon^2}\mathbf{Z}\mathbf{Z}^\top\right)^3\right). \tag{18}$$

For SimSiam in Eqn. (2) and BYOL in Eqn. (3), the losses focus on the trace of $\mathbf{Z}\mathbf{Z}^\top$, which corresponds to the linear term in the Taylor series expansion of $\mathcal{L}_{TCR}$. This makes it a first-order approximation of the TCR loss, emphasizing linear relationships between features.

Notice that we have

$$\text{tr}\left(\mathbf{Z}_1\mathbf{Z}_2^\top\mathbf{Z}_1\mathbf{Z}_2^\top\right) = \sum_{i=1}^{d}\left(\mathbf{Z}_1\mathbf{Z}_2^\top\right)_{ii}^2 + \sum_{i=1}^{d}\sum_{j\neq i}^{d}\left(\mathbf{Z}_1\mathbf{Z}_2^\top\right)_{ij}^2,$$

The second term of Barlow Twins loss in Eqn. (4) is captured in the second-order term of the Taylor expansion of $\mathcal{L}_{\text{TCR}}$, thus making it a second-order approximation. This allows Barlow Twins to capture higher-order statistics of the feature space. For VICReg, the analysis is similar.

The MEC loss defined in Eqn. (8) can be formulated in terms of Matrix Cross-Entropy (MCE) as below:

$$\begin{aligned}
\mathcal{L}_{\text{MEC}} &= \mu\left(\text{MCE}(\mathbf{I}_d, \mathbf{I}_d + \frac{d}{B\epsilon^2}\mathbf{Z}_1\mathbf{Z}_2^\top) - \text{tr}\left(\mathbf{I}_d + \frac{d}{B\epsilon^2}\mathbf{Z}_1\mathbf{Z}_2^\top\right)\right)\\
&= \mu\left(\text{MCE}(\mathbf{I}_d, \mathbf{I}_d + \frac{d}{B\epsilon^2}\mathbf{Z}_1\mathbf{Z}_2^\top) - d - \frac{d}{B\epsilon^2}\text{tr}\left(\mathbf{Z}_1^\top\mathbf{Z}_2\right)\right).
\end{aligned}$$

(19)

Now we see that this formulation based on matrix KL divergence, MCE, and TCR unifies a range of self-supervised learning methods—SimCLR, SimSiam, BYOL, Barlow Twins, VICReg, MCR$^2$, MEC and EMP-SSL—providing a unified perspective that encompasses both **contrastive** and **non-contrastive** paradigms, or in other words, **sample-contrastive** and **dimension-contrastive** methods (Garrido et al., 2022).

# 5 MATRIX INFORMATION-THEORETIC UNIFORMITY AND ALIGNMENT FOR SELF-SUPERVISED LEARNING

From the uniformity principle discussed in Section 4, we aim to ensure that the embeddings have zero mean and a covariance matrix equal to $\frac{1}{d}\mathbf{I}_d$. To align the mean embedding, we can center the embeddings. The following theorem demonstrates that optimizing a matrix uniformity loss $\mathcal{L}_{\text{MCE}}$ can achieve both mean alignment and covariance uniformity in the embeddings.

**Theorem 5.1.** *Let* $\mathbf{x}$ *be a random vector with a distribution supported on the unit hypersphere* $S^{d-1}$. *If the covariance matrix of* $\mathbf{x}$, *denoted by* $\mathbf{C}(\mathbf{x})$, *has the maximal possible effective rank* $d$ *and a trace of at least one, then the expected value of* $\mathbf{x}$ *will be zero, and* $\mathbf{C}(\mathbf{x})$ *will equal* $\frac{1}{d}\mathbf{I}_d$.

To achieve matrix information-theoretic uniformity, we propose the following matrix cross-entropy (MCE) based uniformity loss, where $\mathbf{C}(\mathbf{Z}_1, \mathbf{Z}_2) = \frac{1}{B-1}\mathbf{Z}_1\mathbf{H}_B\mathbf{Z}_2^\top$ represents the sample covariance matrix for simplicity:

$$\begin{aligned}
\mathcal{L}_{\text{Matrix-KL-Uniformity}} = \text{MCE}\left(\frac{1}{d}\mathbf{I}_d, \mathbf{C}\left(\mathbf{Z}_1, \mathbf{Z}_2\right)\right) &= \text{MKL}\left(\frac{1}{d}\mathbf{I}_d\,\|\,\mathbf{C}\left(\mathbf{Z}_1, \mathbf{Z}_2\right)\right) + \text{ME}\left(\frac{1}{d}\mathbf{I}_d\right)\\
&= \text{MKL}\left(\frac{1}{d}\mathbf{I}_d\,\|\,\mathbf{C}\left(\mathbf{Z}_1, \mathbf{Z}_2\right)\right) + \text{Const.}
\end{aligned}$$

(20)

For ease of optimization, a regularization term $\lambda\mathbf{I}_d$ may be added to the cross-covariance to ensure it is non-singular. This adjustment aligns with TCR and MEC methods, differing mainly in mean normalization. An alternative approach is the auto-covariance uniformity loss $\sum_i \text{MCE}\left(\frac{1}{d}\mathbf{I}_d, \mathbf{C}\left(\mathbf{Z}_i, \mathbf{Z}_i\right)\right)$, which is left for future exploration. Interestingly, the first and second terms of the ColorInfomax loss in Eqn. (7) correspond to uncentered auto-covariance MCE.

## 5.1 MATRIX INFORMATION-THEORETIC ALIGNMENT

To directly optimize the alignment of representations in self-supervised learning, we propose the following loss function focusing on the matrix KL divergence between two covariance matrices:

$$\begin{aligned}
\mathcal{L}_{\text{Matrix-KL-Alignment}} &= \text{MKL}\left(\text{C}(\mathbf{Z}_1, \mathbf{Z}_1)\,\|\,\text{C}(\mathbf{Z}_2, \mathbf{Z}_2)\right)\\
&= \text{MCE}\left(\text{C}(\mathbf{Z}_1, \mathbf{Z}_1), \text{C}(\mathbf{Z}_2, \mathbf{Z}_2)\right) - \text{ME}(\text{C}(\mathbf{Z}_1, \mathbf{Z}_1)).
\end{aligned}$$

(21)

Applying the stop gradient technique to the first branch $\mathbf{Z}_1$, as utilized in SimSiam Hua (2021), renders the second term $\mathrm{ME}(\mathrm{C}(\mathbf{Z}_1, \mathbf{Z}_1))$ a constant in the Matrix-KL-Alignment loss. Even without the stop gradient, this term can be integrated into the Matrix-KL-Uniformity loss.

**Lemma 5.2** (Frobenius norm and trace operator)**.**

$$||\mathbf{P} - \mathbf{Q}||_F^2 = ||\mathbf{P}||_F^2 + ||\mathbf{Q}||_F^2 - 2\operatorname{tr}(\mathbf{P}^\top \mathbf{Q}).$$

From Lemma 5.2 and considering the self-adjoint nature of covariance matrices, we observe that various self-supervised learning methods optimize a similar objective involving Frobenius norm, namely SimSiam (Eqn. (2)), BYOL (Eqn. (3)), the third term in VICReg (Eqn. (5)), the last term of MEC (Eqn. (19)), and the third term of the ColorInfoMax loss (Eqn. (7)), also the alignment term in spectral contrastive loss (HaoChen et al., 2021). These methods effectively target $-\|\mathbf{Z}_1 - \mathbf{Z}_2\|_F^2$ (since $\|\mathbf{Z}_1\mathbf{Z}_1^\top\|_F$ and $\|\mathbf{Z}_2\mathbf{Z}_2^\top\|_F$ are constant). This optimization converges to $\mathbf{Z}_1 = \mathbf{Z}_2$, which is also a feasible global minimizer of $\mathcal{L}_{\text{Matrix-KL-Alignment}}$. However, this loss has a broader solution space: $\mathbf{Z}_1\mathbf{Z}_1^\top = \mathbf{Z}_2\mathbf{Z}_2^\top$. In the context of aligning the covariance matrices $\mathbf{Z}_1\mathbf{Z}_1^\top$ and $\mathbf{Z}_2\mathbf{Z}_2^\top$, the key objective is not merely aligning the feature matrices $\mathbf{Z}_1$ and $\mathbf{Z}_2$ themselves but rather ensuring the covariance matrices of $\mathbf{C}(\mathbf{Z}_1, \mathbf{Z}_1)$ and $\mathbf{C}(\mathbf{Z}_2, \mathbf{Z}_2)$ are identical. This alignment maintains the overarching structure of the data representations, accommodating feature space transformations.

This broader solution space in the optimization landscape allows for a more flexible and nuanced approach to learning representations. The goal extends beyond making $\mathbf{Z}_1$ and $\mathbf{Z}_2$ identical, it offers a larger feasible solution space to balance the pursuit of uniformity and alignment.

## 5.2 MATRIX-SSL: UNIFORMITY AND ALIGNMENT

As we have presented an improved loss for uniformity, now generalizing Wang & Isola (2020)'s axiomatic understanding of contrastive learning, we propose the matrix information-theoretic uniformity and alignment framework to improve self-supervised learning:

$$\mathcal{L}_{\text{Matrix-SSL}} = \mathcal{L}_{\text{Matrix-KL-Uniformity}} + \gamma \cdot \mathcal{L}_{\text{Matrix-KL-Alignment}}. \tag{22}$$

## 6 DIMENSIONAL COLLAPSE, EFFECTIVE RANK AND MATRIX ENTROPY

Revisiting Section 4, we established the relationship between dimension-contrastive self-supervised learning and matrix information-theoretic uniformity. This connection leads us to evaluate the extent of "uniformity" (dimensional collapse), we will first discuss the close relationship of effective rank and matrix entropy (and von Neumann entropy).

### 6.1 EFFECTIVE RANK AND RANK INCREASING PHENOMENON

Zhuo et al. (2023) find an intriguing phenomenon that during the optimization course of self-supervised learning, the effective rank of the feature (empirical) covariance matrix consistently increases.

We have presented Proposition 3.6 in Section 3.4 which captures the closed-form relationship among effective rank and matrix information-theoretic quantities. Note the batch auto-correlation matrix is a positive semi-definite matrix with all of its diagonal 1. As we have mentioned earlier, many dimension-contrastive losses can be understood from the matrix information-theoretic uniformity viewpoint. As such, during training the matrix KL divergence minimizes, thus $\frac{1}{B}\mathbf{Z}\mathbf{Z}^\top$ is anticipated to progressively align more with $\frac{1}{d}\mathbf{I}_d$. By the fact that $\frac{1}{d}\mathbf{I}_d$ achieves the maximal possible (matrix) entropy, the rank-increasing phenomenon (Zhuo et al., 2023) can be well understood. **Thus we may treat the effective rank as an exact metric to measure the extent of the dimensional collapse.**

Feature representations acquired through a deep neural network employing a cross-entropy (CE) loss optimized by stochastic gradient descent, are capable of attaining zero loss (Du et al., 2018) with arbitrary label assignments (Zhang et al., 2021). A phenomenon known as neural collapse (NC) (Papyan et al., 2020) is observed when training of the neural network continues beyond zero loss with CE. Based on this, we propose to use effective rank as a tool to investigate the difference between supervised, contrastive, and feature contrastive methods, more details can be found in Appendix E.

# 7 EXPERIMENTS

## 7.1 EXPERIMENTAL SETUP

**Dataset and data augmentations.** We implement our proposed Matrix-SSL method on the self-supervised learning task of ImageNet (Deng et al., 2009) dataset. We use precisely the same data augmentation protocols and hyperparameters with previous baselines such as BYOL (Grill et al., 2020), SimSiam (Chen & He, 2021) and MEC (Liu et al., 2022), etc. In detail, our augmentation protocol consists of random cropping, color jittering, color dropping (grayscale), left-right flipping, Gaussian blurring, and solarization. We augment each image twice to get two different views during each training iteration. For more on experiment details, please refer to Appendix F.

**Pseudo-code.** The pseudo-code for Matrix-SSL is shown as Algorithm 1 in the Appendix.

## 7.2 EVALUATION RESULTS

**Linear evaluation.** We follow the standard linear evaluation protocol (Chen et al., 2020a; Grill et al., 2020; Chen & He, 2021). We freeze the parameters of the backbone encoder and then connect a linear classification layer after it, and train the linear layer in the supervised setting. During training, each image is augmented by random cropping, resizing to 224×224, and random horizontal flipping. At test time, each image is resized to 256×256 and center cropped to 224× 224.

Linear evaluation of the Top-1 accuracy result pre-trained with 100, 200, and 400 epochs on ImageNet (Deng et al., 2009) dataset was shown in Table 1. Notice that we use ResNet50 backbone as default for a fair comparison. Matrix-SSL consistently outperforms baselines across various pre-training epochs.

Table 1: **Linear evaluation** results (Top-1 accuracy) on ImageNet dataset with different pre-training epochs using ResNet50 backbone, **Bold** means the best, underline means the second.

| Method | Pre-training Epochs | | |
|---|---|---|---|
| | 100 | 200 | 400 |
| SimCLR | 66.5 | 68.3 | 69.8 |
| MoCo v2 | 67.4 | 69.9 | 71.0 |
| BYOL | 66.5 | 70.6 | 73.2 |
| SwAV | 66.5 | 69.1 | 70.7 |
| SimSiam | 68.1 | 70.0 | 70.8 |
| Barlow Twins | 67.3 | 70.2 | 71.8 |
| VICReg | 68.6 | – | – |
| MEC | 70.6 | 71.9 | 73.5 |
| Matrix-SSL (Ours) | **71.1** | **72.3** | **73.6** |

**Transfer learning.** Following the common protocol of previous works (Chen et al., 2020b; Chen & He, 2021; Liu et al., 2022), we finetune the pre-trained models on MS-COCO (Lin et al., 2014) object detection and instance segmentation tasks. Table 2 summarizes experiment results of baseline models and Matrix-SSL. The experiment showed that Matrix-SSL consistently outperformed the baselines. It is worth mentioning that Matrix-SSL was only pre-trained for 400 epochs, but it already performed better than all the baselines pre-trained for 800 epochs.

Table 2: **Transfer learning on object detection and instance segmentation tasks. For a fair comparison, we employ a 2-view setting for all methods.** We finetune models pre-trained on ImageNet, with exactly the same experiment settings as SimSiam and MEC. **All baseline models are pre-trained with 800 or 1000 epochs, and our model is pre-trained with 400 epochs.** **Bold** means the best, underline means the second.

| Method | COCO detection | | | COCO instance segmentation | | |
|---|---|---|---|---|---|---|
| | $AP_{50}$ | AP | $AP_{75}$ | $AP_{50}^{mask}$ | $AP^{mask}$ | $AP_{75}^{mask}$ |
| SimCLR | 57.7 | 37.9 | 40.9 | 54.6 | 33.3 | 35.3 |
| MoCo v2 | 58.9 | 39.3 | 42.5 | 55.8 | 34.4 | 36.5 |
| BYOL | 57.8 | 37.9 | 40.9 | 54.3 | 33.2 | 35.0 |
| SwAV | 58.6 | 38.4 | 41.3 | 55.2 | 33.8 | 35.9 |
| Barlow Twins | 59.0 | 39.2 | 42.5 | 56.0 | 34.3 | 36.5 |
| SimSiam | 59.3 | 39.2 | 42.1 | 56.0 | 34.4 | 36.7 |
| VICReg | - | 40.0 | - | - | - | 36.7 |
| MEC | 59.8 | 39.8 | 43.2 | 56.3 | 34.7 | 36.8 |
| Matrix-SSL (400 Epoch) | **60.8** | **41.0** | **44.2** | **57.5** | **35.6** | **38.0** |

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

## A  CONCLUSION

In this paper, we provide a matrix information-theoretic perspective for understanding the self-supervised learning methods. We are confident that our perspective will not only offer a refined and alternative comprehension of self-supervised learning methods but will also act as a catalyst for the design of increasingly robust and effective algorithms in the future.

## B  MATRIX CROSS-ENTROPY FOR LARGE LANGUAGE MODELS

**We further introduce representation learning into the language model regime and use the matrix cross-entropy loss to fine-tune large language models, achieving SOTA results on the GSM8K dataset for mathematical reasoning with 72.3% accuracy using a 7B model, with a margin of 3.1% over standard cross-entropy loss on the GSM8K dataset, and even surpassing the Minerva 540B model.**

Now let us try to pre-train / fine-tune large language models with MCE. Consider the target distribution $\mathbf{p}$ given by the training corpus (which is typically one-hot) and the output distribution $\mathbf{q}$ given by the output of the language model. Suppose we have $l_2$ normalized representation vectors $\mathbf{e_i} \in \mathbb{R}^d$ (column vectors) for tokens $v_i, i \in [n]$, where $n$ is the vocabulary size. One could use LM head embeddings, word embeddings, or any other representation vectors of the models. In our experiments, we use the LM head embeddings as default.

The main intuition behind our method is that the similarity among the representation vector of different words (tokens) can be utilized to address the **synonym phenomenon** and **polysemous phenomenon** within natural language.

For example, "Let **'s** think step by step" should be similar to "Let **us** think step by step". This intricate part hasn't been captured by the classical cross-entropy loss.

For auto-regressive LLMs with tokens $k \in \{1, 2, \cdots, K\}$, we define positive semi-definite matrices $\mathbf{P} \in \mathbb{R}^{d \times d}$ and $\mathbf{Q} \in \mathbb{R}^{d \times d}$ as below:

$$\mathbf{P}^{(k)} = \sum_i \left( p_i^{(k)} \cdot \mathbf{e}_i \mathbf{e}_i^\top \right), \qquad \mathbf{Q}^{(k)} = \sum_j \left( q_j^{(k)} \cdot \mathbf{e}_j \mathbf{e}_j^\top \right).$$

Then we define the following loss as our objective:

$$\begin{aligned}
\mathcal{L}_{\text{Matrix-LLM}} &= \sum_k \text{CE}(\mathbf{p}^{(k)}, \mathbf{q}^{(k)}) + \sum_k \text{MCE}(\mathbf{P}^{(k)}, \mathbf{Q}^{(k)}) \\
&= -\sum_k \sum_i p_i^k \log q_i^{(k)} - \sum_k \text{tr}(\mathbf{P}^{(k)} \log \mathbf{Q}^{(k)}) + \sum_k \text{tr}\left( \mathbf{Q}^{(k)} \right).
\end{aligned} \tag{23}$$

### B.1  EXPERIMENTS ON FINE-TUNING LLMS

**Training Pipeline.**  We use Llemma-7B (Azerbayev et al., 2023) as the base model, which is the CodeLLaMA model continue pretrained on Openwebmath dataset (Paster et al., 2023). We then use $\mathcal{L}_{\text{Matrix-LLM}}$ to fine-tune it on the MetaMath dataset (Yu et al., 2023).

We evaluated the performance of different models on the mathematical reasoning dataset GSM8K (Cobbe et al., 2021) and MATH dataset (Hendrycks et al., 2021), using different loss functions and training methods.

**Experimental Results.**  The results was shown in Table 3. We compared our results against baseline methods, including Minerva (Lewkowycz et al., 2022), WizardMath (Luo et al., 2023), and MetaMath (Yu et al., 2023) + Llemma (Azerbayev et al., 2023) using CE.

| Model | Training method | GSM8K (%) | MATH (%) |
|---|---|---|---|
| Minerva 8B | CE | 16.2 | 14.1 |
| Minerva 62B | CE | 52.4 | 27.6 |
| Minerva 540B | CE | 58.8 | 33.6 |
| WizardMath 7B | RL | 54.9 | 10.7 |
| WizardMath 13B | RL | 63.9 | 14.0 |
| WizardMath 70B | RL | 81.6 | 22.7 |
| LLaMA2 70B | CE | 56.8 | 13.5 |
| MetaMath 7B | CE | 66.5 | 19.8 |
| Llemma 7B | CE | 36.4 | 18.0 |
| Llemma-MetaMath 7B | CE | 69.2 | 30.0 |
| Llemma-MetaMath 7B | CE + MCE | **72.3 (+3.1)** | **30.2 (+0.2)** |

Table 3: Comparison of different models on mathematical reasoning benchmarks.

## ARCHITECTURE OF MATRIX-SSL

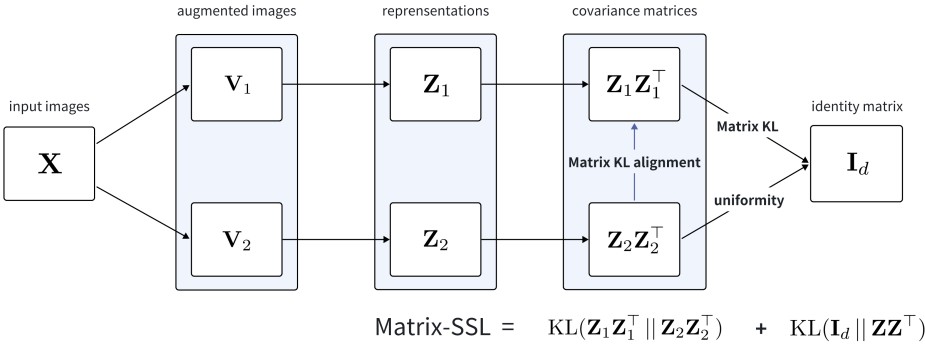

Matrix-SSL $= \mathrm{KL}(\mathbf{Z}_1\mathbf{Z}_1^\top \,\|\, \mathbf{Z}_2\mathbf{Z}_2^\top) + \mathrm{KL}(\mathbf{I}_d \,\|\, \mathbf{Z}\mathbf{Z}^\top)$

Figure 1: Illustration of the architecture of Matrix-SSL, we directly optimize the matrix KL divergence between covariance matrices, achieving higher-order uniformity and alignment.

## PSEUDO-CODE

**Pseudo-code.** The pseudo-code for Matrix-SSL is shown below.

## C   APPENDIX FOR PROOFS

**Proof of Lemma 3.4.**

*Proof.* Consider any non-zero matrix $\mathbf{A} \in \mathbb{R}^{m \times n}$. We want to show that $\mathbf{A}\mathbf{A}^\top$ is positive semi-definite.

Recall that a matrix $\mathbf{B}$ is positive semi-definite if for all vectors $\mathbf{x} \in \mathbb{R}^m$, it holds that $\mathbf{x}^\top \mathbf{B} \mathbf{x} \geq 0$. We will apply this definition to $\mathbf{A}\mathbf{A}^\top$.

Consider any vector $\mathbf{x} \in \mathbb{R}^m$. We compute $\mathbf{x}^\top(\mathbf{A}\mathbf{A}^\top)\mathbf{x}$ as follows:
$$\mathbf{x}^\top(\mathbf{A}\mathbf{A}^\top)\mathbf{x} = (\mathbf{x}^\top \mathbf{A})(\mathbf{A}^\top \mathbf{x})$$
$$= \|\mathbf{A}^\top \mathbf{x}\|^2.$$

---

**Algorithm 1:** PyTorch-style Pseudo-code for Matrix-SSL

---

```
# f:  encoder network
# B:  batch size
# L_Matrix-Uniformity:  Matrix-KL-uniformity loss
# L_Matrix-KL-Alignment:  Matrix-KL-alignment loss
# γ:  weight ratio between Matrix-KL-Uniformity term and
 Matrix-KL-Alignment term
for X in loader:
    # augment a batch of B images in X
    X₁, X₂ = aug(X), aug(X)

    # calculate l₂ normalized embeddings
    Z₁, Z₂ = f(X₁), f(X₂)

    # calculate uniformity and alignment loss
    uniformity_loss = L_Matrix-KL-Uniformity(Z₁, Z₂)
    alignment_loss = L_Matrix-KL-Alignment(Z₁, Z₂)

    # calculate loss
    loss = uniformity_loss + γ * alignment_loss

    # optimization step
    loss.backward()
    optimizer.step()
```

---

The last equality holds because the expression $(\mathbf{x}^\top \mathbf{A})(\mathbf{A}^\top \mathbf{x})$ represents the squared norm of the vector $\mathbf{A}^\top \mathbf{x}$.

Since the squared norm of any vector is always non-negative, $\|\mathbf{A}^\top \mathbf{x}\|^2 \geq 0$ for any $\mathbf{x} \in \mathbb{R}^m$.

Therefore, $\mathbf{x}^\top (\mathbf{A}\mathbf{A}^\top)\mathbf{x} \geq 0$ for all $\mathbf{x} \in \mathbb{R}^m$, which means that $\mathbf{A}\mathbf{A}^\top$ is positive semi-definite.

This completes the proof. $\qquad\square$

**Proof of Proposition 3.6.** Recall the definition of matrix KL divergence:

$$\mathrm{KL}(\mathbf{P} \,\|\, \mathbf{Q}) = \mathrm{tr}(\mathbf{P} \log \mathbf{P} - \mathbf{P} \log \mathbf{Q} - \mathbf{P} + \mathbf{Q}),$$

Substitute $\mathbf{P} = \frac{1}{B}\mathbf{Z}\mathbf{Z}^\top$ and $\mathbf{Q} = \frac{1}{d}\mathbf{I}_d$ into this:

$$
\begin{aligned}
\mathrm{KL}\left(\frac{1}{B}\mathbf{Z}\mathbf{Z}^\top \,\|\, \frac{1}{d}\mathbf{I}_d\right) &= \mathrm{tr}\left(\frac{1}{B}\mathbf{Z}\mathbf{Z}^\top \log\left(\frac{1}{B}\mathbf{Z}\mathbf{Z}^\top\right) - \frac{1}{B}\mathbf{Z}\mathbf{Z}^\top \log\left(\frac{1}{d}\mathbf{I}_d\right) - \frac{1}{B}\mathbf{Z}\mathbf{Z}^\top + \frac{1}{d}\mathbf{I}_d\right) \\
&= \mathrm{tr}\left(\frac{1}{B}\mathbf{Z}\mathbf{Z}^\top \log\left(\frac{1}{B}\mathbf{Z}\mathbf{Z}^\top\right) + \frac{\log d}{B}\mathbf{Z}\mathbf{Z}^\top - \frac{1}{B}\mathbf{Z}\mathbf{Z}^\top + \frac{1}{d}\mathbf{I}_d\right) \\
&= -\mathrm{VNE}\left(\frac{1}{B}\mathbf{Z}\mathbf{Z}^\top\right) + \frac{\log d}{B}\mathrm{tr}(\mathbf{Z}\mathbf{Z}^\top) - \frac{1}{B}\mathrm{tr}(\mathbf{Z}\mathbf{Z}^\top) + \frac{1}{d}\mathrm{tr}(\mathbf{I}_d) \\
&= -\mathrm{VNE}\left(\frac{1}{B}\mathbf{Z}\mathbf{Z}^\top\right) + \log d - 1 + \frac{d}{d} \\
&= -\mathrm{VNE}\left(\frac{1}{B}\mathbf{Z}\mathbf{Z}^\top\right) + \log d,
\end{aligned}
$$

From this, we conclude that:

$$\mathrm{VNE}\left(\frac{1}{B}\mathbf{Z}\mathbf{Z}^\top\right) = -\mathrm{KL}\left(\frac{1}{B}\mathbf{Z}\mathbf{Z}^\top \,\|\, \frac{1}{d}\mathbf{I}_d\right) + \log d.$$

$$\mathrm{ME}\left(\frac{1}{B}\mathbf{Z}\mathbf{Z}^\top\right) = \mathrm{VNE}\left(\frac{1}{B}\mathbf{Z}\mathbf{Z}^\top\right) + \mathrm{tr}\left(\frac{1}{B}\mathbf{Z}\mathbf{Z}^\top\right) = \mathrm{VNE}\left(\frac{1}{B}\mathbf{Z}\mathbf{Z}^\top\right) + 1.$$

The effective rank is defined as:

$$\mathrm{erank}(\mathbf{A}) = \exp\{\mathrm{H}(p_1, p_2, \ldots, p_n)\},$$

If we substitute $\mathbf{A} = \frac{1}{B}\mathbf{Z}\mathbf{Z}^\top$ and given that $\mathrm{VNE}\left(\frac{1}{B}\mathbf{Z}\mathbf{Z}^\top\right)$ is the entropy of the eigenvalue distribution of $\frac{1}{B}\mathbf{Z}\mathbf{Z}^\top$, then we could directly relate $\mathrm{erank}\left(\frac{1}{B}\mathbf{Z}\mathbf{Z}^\top\right)$ and $\mathrm{VNE}\left(\frac{1}{B}\mathbf{Z}\mathbf{Z}^\top\right)$:

$$\mathrm{erank}\left(\frac{1}{B}\mathbf{Z}\mathbf{Z}^\top\right) = \exp\left\{\mathrm{VNE}\left(\frac{1}{B}\mathbf{Z}\mathbf{Z}^\top\right)\right\} = \exp\left\{\mathrm{ME}\left(\frac{1}{B}\mathbf{Z}\mathbf{Z}^\top\right) - 1\right\}.$$

**Proof of Proposition 4.1.**

*Proof.* The proof is straightforward by using eigendecomposition and the minimization property of classical KL divergence between positive measures. Below is an alternative proof using gradient.

First, recall that for two positive semi-definite matrices $\mathbf{P}, \mathbf{Q} \in \mathbb{R}^{n \times n}$, the matrix KL divergence is defined as:

$$\mathrm{KL}(\mathbf{P}\,\|\,\mathbf{Q}) = \mathrm{tr}(\mathbf{P}\log\mathbf{P} - \mathbf{P}\log\mathbf{Q} - \mathbf{P} + \mathbf{Q}).$$

We want to show that:

$$\mathrm{argmin}_{\mathbf{Q}\succ 0}\,\mathrm{KL}(\mathbf{P}\,\|\,\mathbf{Q}) = \mathbf{P}.$$

Consider the constrained optimization problem:

$$\min_{\mathbf{Q}\succ 0}\,\mathrm{KL}(\mathbf{P}\,\|\,\mathbf{Q}).$$

To find the stationary points of $\mathrm{KL}(\mathbf{P}\,\|\,\mathbf{Q})$, we first take the gradient with respect to $\mathbf{Q}$:

$$\nabla_{\mathbf{Q}}\,\mathrm{KL}(\mathbf{P}\,\|\,\mathbf{Q}) = -\mathbf{P}\mathbf{Q}^{-1} + \mathbf{I},$$

where $\mathbf{I}$ is the identity matrix.

Setting the gradient equal to zero and solving for $\mathbf{Q}$:

$$-\mathbf{P}\mathbf{Q}^{-1} + \mathbf{I} = 0 \implies \mathbf{P}\mathbf{Q}^{-1} = \mathbf{I} \implies \mathbf{Q} = \mathbf{P}.$$

To ensure that the solution is indeed a minimum, we need to verify the second-order conditions. The Hessian of $\mathrm{KL}(\mathbf{P}\,\|\,\mathbf{Q})$ with respect to $\mathbf{Q}$ is a bit complicated to compute directly. However, one can verify that the function is convex with respect to $\mathbf{Q}$ by proving that for all $\mathbf{X} \in \mathbb{R}^{n \times n}$:

$$\mathbf{X}^\top \nabla_{\mathbf{Q}}^2\,\mathrm{KL}(\mathbf{P}\,\|\,\mathbf{Q})\mathbf{X} \geq 0.$$

In this case, the function $\mathrm{KL}(\mathbf{P}\,\|\,\mathbf{Q})$ is convex with respect to $\mathbf{Q}$ and hence, $\mathbf{Q} = \mathbf{P}$ is indeed the global minimum.

We have shown that the function $\mathrm{KL}(\mathbf{P}\,\|\,\mathbf{Q})$ has a stationary point at $\mathbf{Q} = \mathbf{P}$ and that this point is a global minimum, thereby proving the proposition:

$$\mathrm{argmin}_{\mathbf{Q}\succ 0}\,\mathrm{KL}(\mathbf{P}\,\|\,\mathbf{Q}) = \mathbf{P}.$$

$\square$

**Proof of Proposition 4.2.**

*Proof.* The matrix cross-entropy is defined as:

$$\text{MCE}(\mathbf{P}, \mathbf{Q}) = \text{tr}(-\mathbf{P} \log \mathbf{Q} + \mathbf{Q}).$$

Taking the derivative of MCE with respect to $\mathbf{Q}$ :

$$\frac{\partial \text{MCE}}{\partial \mathbf{Q}} = -\mathbf{P}\mathbf{Q}^{-1} + \mathbf{I}.$$

where we used the fact that the derivative of $\log \mathbf{Q}$ with respect to $\mathbf{Q}$ is $\mathbf{Q}^{-1}$.

Setting the derivative to zero gives:

$$-\mathbf{P}\mathbf{Q}^{-1} + \mathbf{I} = \mathbf{0} \implies \mathbf{P}\mathbf{Q}\mathbf{Q}^{-1} = \mathbf{I} \implies \mathbf{P} = \mathbf{Q}.$$

Thus,

$$\text{argmin}_{\mathbf{Q} \succ 0} \text{MCE}(\mathbf{P}, \mathbf{Q}) = \mathbf{P}.$$

$\square$

**Proof of Propostion 4.3.** We directly utilize the notations from the renowned work by Balestriero & LeCun (2022).

*Proof.* The SimCLR loss (Chen et al., 2020a) firstly produces an estimated relation matrix $\widehat{\mathbf{G}}(\mathbf{Z})$ generally using the cosine similarity $(\text{CoSim})$ via

$$(\widehat{\mathbf{G}}(\mathbf{Z}))_{i,j} = \frac{e^{\text{CoSim}(\mathbf{z}_i, \mathbf{z}_j)/\tau}}{\sum_{j=1, j \neq i}^{2B} e^{\text{CoSim}(\mathbf{z}_i, \mathbf{z}_j)/\tau}},$$

with $\tau > 0$ a temperature parameter. Then SimCLR encourages the elements of $\widehat{\mathbf{G}}(\mathbf{Z})$ and $\mathbf{G}$ to match. The most popular solution to achieve that is to leverage the InfoNCE loss given by

$$\mathcal{L}_{\text{SimCLR}} = -\underbrace{\sum_{i=1}^{2B} \sum_{j=1}^{2B} (\mathbf{G})_{i,j} \log(\widehat{\mathbf{G}}(\mathbf{Z}))_{i,j}}_{\text{cross-entropy between matrices}}.$$

If the matrix $\mathbf{G}$ and $\widehat{\mathbf{G}}(\mathbf{Z})$ are both diagonal, then

$$\mathcal{L}_{\text{SimCLR}}^{\text{diag}} = -\sum_{i=1}^{2B} (\mathbf{G})_{i,i} \log \widehat{\mathbf{G}}(\mathbf{Z})_{i,i} = -\text{tr}((\mathbf{G}) \log \widehat{\mathbf{G}}(\mathbf{Z})). \tag{24}$$

$\square$

For more on the element-wise cross-entropy interpretation of SimCLR and its connection to MCE when the matrix is diagonal, please refer to Tan et al. (2023); Zhang et al. (2023).

Let's consider $p_{\text{x}}$, the probability distribution on $\mathcal{X}$ that describes the data distribution. We adopt the standard notation $q = p_{\text{x}} \circ \boldsymbol{f}^{-1}$ as the push-forward measure of $p_{\text{x}}$ under the feature mapping $\boldsymbol{f}$ parameterized by a neural network (note that distribution $q$ is supported on $\boldsymbol{f}(\mathcal{X})$). Given $\|\boldsymbol{f}(x)\|^2 = 1$, it follows that $\boldsymbol{f}(\mathcal{X}) \subseteq S^{d-1}$ (where $S^{d-1}$ is the $d$-dimensional hypersphere with radius 1 ).

**Lemma C.1.** *Let's represent the uniform distribution on $S^{d-1}$. Then the expected value of $\sigma$ is $\mathbf{0}$ and the auto-correlation matrix of $\sigma$ is $\frac{1}{d}\mathbf{I}_d$.*

Applying the inverse image rule, we know that the auto-correlation matrix of $q$ is the covariance of $p_x$. Thus based on the uniformity principle (Wang & Isola, 2020), we want the (empirical) covariance of $q$ to align with the covariance of $\sigma$.

**Proof of Lemma C.1.**

*Proof.* We aim to prove two aspects of the uniform distribution $\sigma$ on the hypersphere $S^{d-1}$: the expected value of $\sigma$ is 0, and the covariance (or auto-correlation matrix) of $\sigma$ is $\frac{1}{d}\mathbf{I}_d$, where $\mathbf{I}_d$ is the $d$-dimensional identity matrix.

The uniform distribution over $S^{d-1}$ implies that all directions on the hypersphere are equally probable. Given the symmetry of $S^{d-1}$, for every point $x$ on the hypersphere, there is an antipodal point $-x$, both equally probable under $\sigma$. Therefore, when integrating over $S^{d-1}$, the contribution of each component of every point is negated by its antipodal counterpart. Thus, the expected value of each component is 0, leading to the expected value of $\sigma$ being the zero vector.

Secondly, the covariance matrix of $\sigma$ is defined as:

$$\mathbb{E}[\mathbf{X}\mathbf{X}^\top] - \mathbb{E}[\mathbf{X}]\mathbb{E}[\mathbf{X}]^\top.$$

Given that $\mathbb{E}[\mathbf{X}]$ is the zero vector, we are left with $\mathbf{X}\mathbf{X}^\top$. The symmetry of the uniform distribution over $S^{d-1}$ implies that for the covariance matrix, the off-diagonal elements (representing covariances between different components) average to zero. The diagonal elements of $\mathbf{X}\mathbf{X}^\top$ represent the squared components of $X$, which, due to the uniformity and symmetry, all have the same expected value. Since $X$ lies on $S^{d-1}$, the sum of the squares of its components equals 1. Hence, the expected value of each squared component is $\frac{1}{d}$. Therefore, the covariance matrix, which is $\mathbb{E}[\mathbf{X}\mathbf{X}^\top]$, becomes $\frac{1}{d}\mathbf{I}_d$.

Thus, the lemma is proved. $\qquad\qquad\qquad\qquad\qquad\qquad\qquad\qquad\qquad\qquad\qquad\qquad\qquad\qquad\square$

**Proof of Theorem 4.4.**

*Proof.* First, begin with $\mathcal{L}_{\text{UMCE}}$ :

$$\mathcal{L}_{\text{UMCE}} = \text{MCE}\left(\frac{1}{d}\mathbf{I}_d + \lambda\mathbf{I}_d, \frac{1}{B}\mathbf{Z}\mathbf{Z}^\top + \lambda\mathbf{I}_d\right),$$

Using the definition of MCE, we get:

$$\mathcal{L}_{\text{UMCE}} = \text{tr}\left(-\left(\frac{1}{d}\mathbf{I}_d + \lambda\mathbf{I}_d\right)\log\left(\frac{1}{B}\mathbf{Z}\mathbf{Z}^\top + \lambda\mathbf{I}_d\right) + \frac{1}{B}\mathbf{Z}\mathbf{Z}^\top + \lambda\mathbf{I}_d\right),$$

Now, let us divide and multiply by $\lambda$ of the term $-\log\left(\frac{1}{B}\mathbf{Z}\mathbf{Z}^\top + \lambda\mathbf{I}_d\right)$:

$$-\log\left(\frac{1}{B}\mathbf{Z}\mathbf{Z}^\top + \frac{\epsilon^2}{d}\mathbf{I}_d\right) = -\log\left(\lambda\left(\frac{1}{\lambda B}\mathbf{Z}\mathbf{Z}^\top + \mathbf{I}_d\right)\right),$$

Now, factor out $\lambda$:

$$-\log\left(\lambda\left(\frac{1}{\lambda B}\mathbf{Z}\mathbf{Z}^\top + \mathbf{I}_d\right)\right) = -\log(\lambda)\mathbf{I}_d - \log\left(\frac{1}{\lambda B}\mathbf{Z}\mathbf{Z}^\top + \mathbf{I}_d\right),$$

Since $\mathcal{L}_{\text{TCR}} = \frac{1}{2}\log\det\left(\mathbf{I}_d + \frac{d}{B\epsilon^2}\mathbf{Z}\mathbf{Z}^\top\right)$, we can rewrite this term in the form of $\mathcal{L}_{\text{TCR}}$.

$$\text{tr}\left(-\log\left(\frac{1}{\lambda B}\mathbf{Z}\mathbf{Z}^\top + \mathbf{I}_d\right)\right) = \text{tr}\left(-\log\left(\mathbf{I}_d + \frac{d}{B\epsilon^2}\mathbf{Z}\mathbf{Z}^\top\right)\right) = 2\mathcal{L}_{\text{TCR}},$$

Upon substitution, it becomes:

$$\mathcal{L}_{\text{UMCE}} = -\text{tr}\left(\left(\frac{1}{d}\mathbf{I}_d + \lambda\mathbf{I}_d\right)(\log(\lambda)\mathbf{I}_d)\right) + 2(1 + d\lambda)\mathcal{L}_{\text{TCR}} + \text{tr}\left(\frac{1}{B}\mathbf{Z}\mathbf{Z}^\top + \lambda\mathbf{I}_d\right),$$

Simplifying, we get:

$$\mathcal{L}_{\text{UMCE}} = -(1 + d\lambda)\log\lambda + 2(1 + d\lambda)\mathcal{L}_{\text{TCR}} + 1 + d\lambda$$
$$= (1 + d\lambda)\left(-\log\lambda + 1 + 2\mathcal{L}_{\text{TCR}}\right).$$

This matches the expression given in the proposition for $\mathcal{L}_{\text{UMCE}}$.

For $\mathcal{L}_{\text{UMKL}}$, Using the definition of Matrix KL divergence, we have:

$$\mathcal{L}_{\text{UMKL}} = \text{KL}\left(\frac{1}{d}\mathbf{I}_d + \lambda\mathbf{I}_d \,\middle|\middle|\, \frac{1}{B}\mathbf{Z}\mathbf{Z}^\top + \lambda\mathbf{I}_d\right),$$

$$= \text{MCE}\left(\frac{1}{d}\mathbf{I}_d + \lambda\mathbf{I}_d, \frac{1}{B}\mathbf{Z}\mathbf{Z}^\top + \lambda\mathbf{I}_d\right) + \text{tr}\left(\mathbf{P}\log\mathbf{P} - \mathbf{P}\right),$$

where $\mathbf{P}$ denotes $\frac{1}{d}\mathbf{I}_d + \lambda\mathbf{I}_d$.

Now, we simplify $\text{tr}\left(\mathbf{P}\log\mathbf{P} - \mathbf{P}\right)$. We know that $\mathbf{P} = \frac{1}{d}\mathbf{I}_d + \lambda\mathbf{I}_d = \left(\frac{1}{d} + \lambda\right)\mathbf{I}_d$.

Since $\mathbf{P}$ is a diagonal matrix with all diagonal entries being $\frac{1}{d} + \lambda$, its matrix logarithm $\log\mathbf{P}$ will also be a diagonal matrix with all diagonal entries being $\log\left(\frac{1}{d} + \lambda\right)$.

Thus, $\text{tr}\left(\mathbf{P}\log\mathbf{P} - \mathbf{P}\right)$ can be simplified as follows:

$$\text{tr}\left(\mathbf{P}\log\mathbf{P} - \mathbf{P}\right) = \text{tr}\left(\left(\frac{1}{d} + \lambda\right)\mathbf{I}_d\left(\log\left(\frac{1}{d} + \lambda\right)\mathbf{I}_d\right) - \left(\frac{1}{d} + \lambda\right)\mathbf{I}_d\right),$$

Since the diagonal matrix $\mathbf{I}_d$ has $d$ ones along its diagonal, the trace operation essentially multiplies each term by $d$. Therefore, we can write:

$$\text{tr}(\mathbf{P}\log\mathbf{P} - \mathbf{P}) = d\left(\left(\frac{1}{d} + \lambda\right)\log\left(\frac{1}{d} + \lambda\right) - \left(\frac{1}{d} + \lambda\right)\right),$$

Further simplifying, we get:

$$\text{tr}(\mathbf{P}\log\mathbf{P} - \mathbf{P}) = d\left(\frac{1}{d} + \lambda\right)\log\left(\frac{1}{d} + \lambda\right) - d\left(\frac{1}{d} + \lambda\right)$$

$$= (1 + d\lambda)(\log(1 + d\lambda) - \log d - 1),$$

Now, we can rewrite $\mathcal{L}_{\text{UMKL}}$ using this result:

$$\begin{aligned}
\mathcal{L}_{\text{UMKL}} &= \mathcal{L}_{\text{UMCE}} + \text{tr}\left(\mathbf{P}\log\mathbf{P} - \mathbf{P}\right) \\
&= \mathcal{L}_{\text{UMCE}} + (1 + d\lambda)(\log(1 + d\lambda) - \log d - 1) \\
&= -(1 + d\lambda)\log\lambda + 2(1 + d\lambda)\mathcal{L}_{\text{TCR}} + 1 + d\lambda + (1 + d\lambda)(\log(1 + d\lambda) - \log d - 1) \\
&= -(1 + d\lambda)\log\lambda + 2(1 + d\lambda)\mathcal{L}_{\text{TCR}} + (1 + d\lambda)\log(1 + d\lambda) - (1 + d\lambda)\log d \\
&= (1 + d\lambda)(-\log\lambda + 2\mathcal{L}_{\text{TCR}} + \log(1 + d\lambda) - \log d) \\
&= (1 + d\lambda)(\log\frac{1 + d\lambda}{\lambda d} + 2\mathcal{L}_{\text{TCR}}).
\end{aligned}$$

This equation represents $\mathcal{L}_{\text{UMKL}}$ in terms of $\mathcal{L}_{\text{TCR}}$ and other constants $d$, $\lambda$, and $B$, thus fulfilling the proposition. $\qquad\square$

**Proof of Theorem 4.5.**

*Proof.* Here we present an alternative proof without resorting to other literature. To prove the theorem, we examine the form of the TCR loss:

$$\mathcal{L}_{\text{TCR}} = -\frac{1}{2}\log\det\left(\mathbf{I}_d + \frac{d}{B\epsilon^2}\mathbf{Z}\mathbf{Z}^\top\right),$$

where $\mathbf{Z} = [\boldsymbol{f}(x_1), \cdots, \boldsymbol{f}(x_B)] \in \mathbb{R}^{d \times B}$.

We note that $\mathbf{Z}\mathbf{Z}^\top$ is a positive semi-definite matrix, as it is the product of a matrix and its transpose. Hence, all its eigenvalues are non-negative. Let these eigenvalues be denoted by $\lambda_1, \lambda_2, \ldots, \lambda_d$.

The determinant of $\mathbf{I}_d + \frac{d}{B\epsilon^2}\mathbf{Z}\mathbf{Z}^\top$ can then be expressed as the product of its eigenvalues:

$$\det\left(\mathbf{I}_d + \frac{d}{B\epsilon^2}\mathbf{Z}\mathbf{Z}^\top\right) = \prod_{i=1}^{d}(1 + \frac{d}{B\epsilon^2}\lambda_i).$$

Since logarithm is a monotonically increasing function, minimizing $\mathcal{L}_{\text{TCR}}$ is equivalent to maximizing the product of $(1 + \frac{d}{B\epsilon^2}\lambda_i)$ terms.

Applying the arithmetic mean-geometric mean inequality, we find that the product of the eigenvalues (and thus the determinant) is maximized when all eigenvalues are equal, i.e., $\lambda_i = \frac{B}{d}$ for all $i$. Therefore, the matrix that maximizes this determinant under the given constraints is one where all eigenvalues are $\frac{B}{d}$.

Hence, the global and unique minimizer of the TCR loss under the constraint $\|\mathbf{z}_i\|_2^2 = 1$ is achieved when $\frac{1}{B}\mathbf{Z}\mathbf{Z}^\top$ has eigenvalues equal to $\frac{1}{d}$, which corresponds to $\frac{1}{B}\mathbf{Z}\mathbf{Z}^\top = \frac{1}{d}\mathbf{I}_d$. □

**Proof of Proposition 4.6.**

*Proof.* Utilizing the property $\mathrm{tr}(\log(\mathbf{A})) = \log(\det(\mathbf{A}))$, the TCR loss function can be rewritten as

$$\mathcal{L}_{\text{TCR}} = -\frac{1}{2}\,\mathrm{tr}\left(\log\left(\mathbf{I}_d + \frac{d}{B\epsilon^2}\mathbf{Z}\mathbf{Z}^\top\right)\right).$$

The Taylor series expansion of the trace log function around $\mathbf{X} = \mathbf{0}$ is

$$\mathrm{tr}(\log(\mathbf{I}_d + \mathbf{X})) \approx \mathrm{tr}(\mathbf{X}) - \frac{1}{2}\,\mathrm{tr}(\mathbf{X}^2) + O(\mathbf{X}^3).$$

Substituting this into $\mathcal{L}_{\text{TCR}}$ yields

$$\mathcal{L}_{\text{TCR}} \approx -\frac{1}{2}\left(\mathrm{tr}\left(\frac{d}{B\epsilon^2}\mathbf{Z}\mathbf{Z}^\top\right) - \frac{1}{2}\,\mathrm{tr}\left(\left(\frac{d}{B\epsilon^2}\mathbf{Z}\mathbf{Z}^\top\right)^2\right)\right) + O\left(\left(\frac{d}{B\epsilon^2}\mathbf{Z}\mathbf{Z}^\top\right)^3\right)$$

$$= -\frac{d}{2B\epsilon^2}\,\mathrm{tr}(\mathbf{Z}\mathbf{Z}^\top) + \frac{d^2}{4B^2\epsilon^4}\,\mathrm{tr}(\mathbf{Z}\mathbf{Z}^\top\mathbf{Z}\mathbf{Z}^\top) + O\left(\left(\frac{d}{B\epsilon^2}\mathbf{Z}\mathbf{Z}^\top\right)^3\right).$$

This concludes the proof. □

**Proof of Theorem 5.1.**

*Proof.* Based on the definition of effective rank presented in Section 3.4, a maximal effective rank of $d$ implies that the covariance matrix has $d$ non-negligible eigenvalues.

Let $\mathbf{x} = [x_1, x_2, \ldots, x_d]^\top$ be a random vector on $S^{d-1}$. The covariance matrix $\mathbf{C}(\mathbf{x})$ of $\mathbf{x}$ is defined as $\mathbb{E}[\mathbf{x}\mathbf{x}^\top] - \mathbb{E}[\mathbf{x}]\mathbb{E}[\mathbf{x}]^\top$.

Since $\mathbf{x}$ is on $S^{d-1}$, $\|\mathbf{x}\|^2 = 1$ for each instance of $\mathbf{x}$, and thus $\mathbb{E}[\mathbf{x}\mathbf{x}^\top]$ is a diagonal matrix with each diagonal element being the expectation of the squared components of $\mathbf{x}$.

The trace of $\mathbf{C}(\mathbf{x})$, which is the sum of its eigenvalues, must be at least 1. Given the maximal effective rank $d$, each of these $d$ eigenvalues must be equal (denote this common value as $\lambda$), resulting in $\mathbf{C}(\mathbf{x}) = \lambda\mathbf{I}_d$.

As $\mathbf{x}$ is distributed on $S^{d-1}$, the expectation $\mathbb{E}[\mathbf{x}]$ is the zero vector due to the symmetry of the distribution over the hypersphere.

Combining Assertions 2 and 3, we find that $\mathbb{E}[\mathbf{x}\mathbf{x}^\top] = \lambda\mathbf{I}_d$. The trace of this matrix, which is $d\lambda$, must be equal to 1, implying $\lambda = \frac{1}{d}$.

Thus, we conclude that if the covariance matrix of $\mathbf{x}$ has the maximal possible effective rank of $d$ and its trace is at least one, then the expected value of $\mathbf{x}$ is zero, and the covariance matrix $\mathbf{C}(\mathbf{x})$ is $\frac{1}{d}\mathbf{I}_d$. □

**Proof of Lemma 5.2.** Lemma 5.2 states the relationship between the Frobenius norm of the difference between two matrices and their individual Frobenius norms and trace operator. The Frobenius norm of a matrix is defined as the square root of the sum of the absolute squares of its elements, which can also be expressed in terms of the trace of a matrix.

*Proof.* Given two matrices $\mathbf{P}$ and $\mathbf{Q}$, the Frobenius norm of a matrix $\mathbf{A}$ is defined as $||\mathbf{A}||_F = \sqrt{\operatorname{tr}(\mathbf{A}^\top \mathbf{A})}$, where $\operatorname{tr}$ denotes the trace of a matrix.

We need to prove that:

$$||\mathbf{P} - \mathbf{Q}||_F^2 = ||\mathbf{P}||_F^2 + ||\mathbf{Q}||_F^2 - 2\operatorname{tr}(\mathbf{P}^\top \mathbf{Q}).$$

Starting from the left-hand side:

$$
\begin{aligned}
||\mathbf{P} - \mathbf{Q}||_F^2 &= \operatorname{tr}((\mathbf{P} - \mathbf{Q})^\top (\mathbf{P} - \mathbf{Q})) \\
&= \operatorname{tr}(\mathbf{P}^\top \mathbf{P} - \mathbf{P}^\top \mathbf{Q} - \mathbf{Q}^\top \mathbf{P} + \mathbf{Q}^\top \mathbf{Q}) \\
&= \operatorname{tr}(\mathbf{P}^\top \mathbf{P}) - \operatorname{tr}(\mathbf{P}^\top \mathbf{Q}) - \operatorname{tr}(\mathbf{Q}^\top \mathbf{P}) + \operatorname{tr}(\mathbf{Q}^\top \mathbf{Q}) \\
&= ||\mathbf{P}||_F^2 + ||\mathbf{Q}||_F^2 - 2\operatorname{tr}(\mathbf{P}^\top \mathbf{Q}),
\end{aligned}
$$

In the third step, we used the linearity of the trace operator and the fact that the trace of a transpose is the same as the trace of the matrix itself. This completes the proof. $\square$

**Lemma C.2.** *Let $\mathbf{Z}_1, \mathbf{Z}_2 \in \mathbb{R}^{d \times B}$ where $d$ is the dimensionality of the data and $B$ is the number of samples. The cross-covariance matrix $\mathbf{C}(\mathbf{Z}_1, \mathbf{Z}_2)$ can be expressed as:*

$$\mathbf{C}(\mathbf{Z}_1, \mathbf{Z}_2) = \frac{1}{B-1} \mathbf{Z}_1 \mathbf{H}_B \mathbf{Z}_2^\top,$$

*where $\mathbf{H}_B = \mathbf{I}_B - \frac{1}{B} \mathbf{1_B} \mathbf{1_B}^\top$ is the centering matrix.*

**Proof of Lemma C.2.**

*Proof.* To prove the lemma, we first apply the centering matrix $\mathbf{H}_B$ to $\mathbf{Z}_1$ and $\mathbf{Z}_2$ as follows:

$$
\begin{aligned}
\bar{\mathbf{Z}}_1 &= \mathbf{Z}_1 \mathbf{H}_B, \\
\bar{\mathbf{Z}}_2 &= \mathbf{Z}_2 \mathbf{H}_B.
\end{aligned}
$$

These equations remove the mean of each row, effectively centering the data.

The cross-covariance matrix for the centered data $\bar{\mathbf{Z}}_1$ and $\bar{\mathbf{Z}}_2$ is then given by:

$$\mathbf{C}(\bar{\mathbf{Z}}_1, \bar{\mathbf{Z}}_2) = \frac{1}{B-1} \bar{\mathbf{Z}}_1 \bar{\mathbf{Z}}_2^\top.$$

Substituting the expressions for $\bar{\mathbf{Z}}_1$ and $\bar{\mathbf{Z}}_2$, we get:

$$\mathbf{C}(\mathbf{Z}_1, \mathbf{Z}_2) = \frac{1}{B-1} (\mathbf{Z}_1 \mathbf{H}_B)(\mathbf{Z}_2 \mathbf{H}_B)^\top.$$

Because $\mathbf{H}_B$ is symmetric ($\mathbf{H}_B = \mathbf{H}_B^\top$) and idempotent ($\mathbf{H}_B^2 = \mathbf{H}_B$), this expression simplifies to:

$$\mathbf{C}(\mathbf{Z}_1, \mathbf{Z}_2) = \frac{1}{B-1} \mathbf{Z}_1 \mathbf{H}_B \mathbf{Z}_2^\top,$$

completing the proof. $\square$

## D  NEURAL COLLAPSE AND DIMENSIONAL COLLAPSE

Feature representations acquired through a deep neural network employing a cross-entropy (CE) loss optimized by stochastic gradient descent, are capable of attaining zero loss (Du et al., 2018) with arbitrary label assignments (Zhang et al., 2021). A phenomenon which known as neural collapse (NC) (Papyan et al., 2020) is observed when training of the neural network continues beyond zero loss with CE. Galanti et al. (2021) demonstrate that the NC phenomenon can facilitate some transfer learning tasks. However, potential concerns associated with neural collapse exist, as Ma et al. (2023) posit that the total within-class features collapse may not be ideal for fine-grained classification tasks.

The NC phenomenon embodies the following characteristics (Zhu et al., 2021):

- Variability collapse: The intra-class variability of the final layer's features collapse to zero, signifying that all the features of a single class concentrate on the mean of these features for each class respectively.

- Convergence to Simplex ETF: Once centered at their global mean, the class-means are simultaneously linearly separable and maximally distant on a hypersphere. This results in the class-means forming a simplex equiangular tight frame (ETF), a symmetrical structure determined by a set of points on a hypersphere that is maximally distant and equiangular to each other.

- Convergence to self-duality: The linear classifiers, existing in the dual vector space of the class-means, converge to their respective class-mean and also construct a simplex ETF.

- Simplification to Nearest Class-Center (NCC): The linear classifiers behaviors similarly to the nearest class-mean decision rule.

Here we present the definition of standard $K$-Simplex ETF and general $K$-Simplex ETF (Papyan et al., 2020).

**Definition D.1** ($K$-Simplex ETF). A standard Simplex ETF is characterized as a set of points in $\mathbb{R}^K$, defined by the columns of

$$\mathbf{M} = \sqrt{\frac{K}{K-1}} \left( \mathbf{I}_K - \frac{1}{K} \mathbf{1}_K \mathbf{1}_K^\top \right),$$

where $\mathbf{I}_K \in \mathbb{R}^{K \times K}$ is the identity matrix, and $\mathbf{1}_K \in \mathbb{R}^K$ represents a all-one vector. Consequently, we also obtain

$$\mathbf{M}^\top \mathbf{M} = \mathbf{M}\mathbf{M}^\top = \frac{K}{K-1} \left( \mathbf{I}_K - \frac{1}{K} \mathbf{1}_K \mathbf{1}_K^\top \right).$$

**Definition D.2** (General $K$-Simplex ETF). A general Simplex ETF is characterized as a set of points in $\mathbb{R}^K$, defined by the columns of

$$\tilde{\mathbf{M}} = \alpha \mathbf{U} \mathbf{M},$$

where $\alpha \in \mathbb{R}_+$ is a scale factor, and $\mathbf{U} \in \mathbb{R}^{p \times K}$ ($p \geq K$) is a partial orthogonal matrix $\mathbf{U}^\top \mathbf{U} = \mathbf{I}$.

Zhu et al. (2021) further studied the problem using an unconstrained feature model that separates the topmost layers from the classifier of the neural network. They established that the conventional cross-entropy loss with weight decay presents a benign global landscape, where the only global minimizers are the Simplex ETFs and all other critical points are strict saddles exhibiting negative curvature directions.

The study was later extended (Zhou et al., 2022), demonstrating through a global solution and landscape analysis that a wide range of loss functions, including commonly used label smoothing (LS) and focal loss (FL), display Neural Collapse. Therefore, all pertinent losses (i.e., CE, LS, FL, MSE) yield comparable features on training data.

## E MEASURING DIMENSIONAL COLLAPSE

Papyan et al. (2020) discuss the fascinating occurrence of neural collapse during the training of a supervised neural network utilizing cross-entropy loss for classification tasks. Contrastive learning may have similar effects of dimensional collapse due to its spectral clustering nature (Tan et al., 2023). As dimension-contrastive learning can be seen as pursuing uniformity, we are also interested in discovering the relationship between dimension-contrastive learning and dimensional collapse.

Figure 2 illustrates that the non-contrastive method, Barlow Twins, exhibits greater intra-class variability than the contrastive method, SimCLR. However, for larger samples and classes (e.g., Figure 5 in Appendix F), this observation is qualitative explicit. To quantify this observation, we propose the introduction of metrics involving class-specific information to quantify dimensional collapse. These measures may enhance our understanding of the differences among supervised learning, contrastive, and non-contrastive SSL.

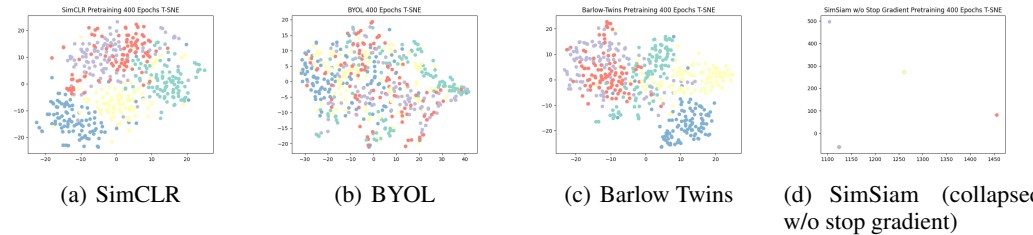

(a) SimCLR     (b) BYOL     (c) Barlow Twins     (d) SimSiam (collapsed w/o stop gradient)

Figure 2: Visualization of feature representation for images in 5 different classes from CIFAR-100 dataset via t-SNE of various self-supervised learning methods. We find that SimCLR has larger inter-class variability than others, as the clusters seem more separable. For illustration, we also introduce a collapsed representation via SimSiam without stop gradient operation.

Assuming a total of $K$ classes and $n$ labeled samples $\{x_i, y_i\}_{i=1}^n$, denote the number of samples in each class $c$ as $n_c$, i.e., $n_c = |\{i \mid y_i = c\}|$. We define the *intra-class effective rank* and *inter-class effective rank* as follows.

**Definition E.1** (Intra-class effective rank)**.** Denote the class-mean vector of each class $c$ as $\mu_c = \frac{1}{n_c} \sum_{y_i=c} \mathbf{f}(\mathbf{x}_i)$, and denote $\mathbf{C}(\mathbf{f}(x) \mid y)) = \frac{1}{n_y} \sum_{y_i=y} (\mathbf{f}(x_i) - \mu_y)(\mathbf{f}(x_i) - \mu_y)^\top$. We define *intra-class effective rank* (intra-class erank) as

$$\operatorname{erank}_{\text{intra-class}} \triangleq \frac{1}{K} \sum_{y \in [K]} \operatorname{erank}(\mathbf{C}(\mathbf{f}(\mathbf{x}) \mid y))), \tag{25}$$

which can be viewed as an empirical approximation of $\mathbb{E}_{y \in [K]} [\operatorname{erank}(\mathbf{C}(\mathbf{f}(x) \mid y))]$, where $x$ is drawn from $p_{\text{data}}$.

**Definition E.2** (Inter-class effective rank)**.** Denote global mean of representation as $\mu_G = \frac{1}{n} \sum_{i \in [n]} \mathbf{f}(x_i)$, then we define *inter-class effective rank* (inter-class erank) as the effective rank of the covariance matrix of all $C$ class-mean vectors,

$$\operatorname{erank}_{\text{inter-class}} \triangleq \operatorname{erank}[\frac{1}{K} \sum_{i \in [K]} (\mu_i - \mu_G)(\mu_i - \mu_G)^\top]. \tag{26}$$

When class are balanced, intra-class erank is approximately $\operatorname{erank}(\mathbf{C}_{y \in [K]}(E[\mathbf{f}(x) \mid y]))$, where $x$ is drawn from $p_{\text{data}}$.

**Remark.** These two metrics can be interpreted as an effective rank factorization of the two terms in the total covariance theorem.

From illustrative examples shown in Figure 3, we observe that SimCLR, as a contrastive method, exhibits a consistent decrease in intra-class effective rank during training. This empirical evidence corroborates the spectral clustering interpretation of contrastive learning. On the contrary, non-contrastive methods like BYOL and Barlow Twins, owing to the inherent property of kernel-uniformity loss (and its low-order Taylor approximations) tending towards a uniform distribution, exhibit larger intra-class effective ranks that continue to increase during training. Regarding the inter-class effective rank, a metric for global class-means effective rank, all three methods show a consistent increase.

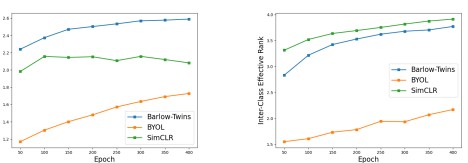

(a) Intra-class erank on test dataset     (b) Inter-class erank on test dataset

Figure 3: Intra-class effective rank and inter-class effective rank. It is obvious that intra-class effective rank continues to grow for BYOL or Barlow Twins, but not for SimCLR.

We now present some theoretical properties of effective rank and its connections to an equiangular tight frame (ETF). The following theorem suggests that a larger effective rank of the Gram matrix is beneficial for expressiveness.

**Theorem E.3** (Maximize effective rank forms a equiangular tight frame (ETF))**.** *For $K$ vectors $\mathbf{z}_i$ ($1 \leq i \leq K$), each lying on $S^{d-1}$. Assuming the latent dimension $d$ satisfies $d \geq K$ and the mean of $\mathbf{z}_i$ is $\mathbf{0}$, denote $\mathbf{Z} = [\mathbf{z}_1, \cdots, \mathbf{z}_K]$. If the Gram matrix $\mathbf{Z}^\top \mathbf{Z}$ has an effective rank of $K - 1$, it implies the existence of an equiangular tight frame (ETF) in the orthonormal span of $\mathbf{z}_i$. Conversely, the Gram matrix of any ETF has an effective rank of $K - 1$.*

*Proof.* Since the mean vector is $\mathbf{0}$, the Gram matrix can have an effective rank of at most $K - 1$. By Property 1 in Roy & Vetterli (2007), we deduce that the Gram matrix $\mathbf{Z}^\top \mathbf{Z}$ has $K - 1$ equal eigenvalues and one eigenvalue equal to $0$.

The trace of the Gram matrix equals $K$ because $\mathbf{z}_i$ lies on $S^{d-1}$. Hence, the Gram matrix has $K - 1$ equal eigenvalues of $\frac{K}{K-1}$ and one eigenvalue of $0$. Therefore, the Gram matrix shares the same eigenvalues (spectrum) as $\frac{K}{K-1} \mathbf{H}_K$, where $\mathbf{H}_K$ is the centering matrix $\mathbf{I}_K - \frac{1}{K} \mathbf{1_K} \mathbf{1_K}^\top$.

Given the orthonormal standard form, there exists an orthonormal matrix $\mathbf{Q} \in \mathbb{R}^{K \times K}$ such that $\mathbf{Q}^\top (\mathbf{Z}^\top \mathbf{Z}) \mathbf{Q} = \frac{K}{K-1} \mathbf{H}_K$. According to Lemma 11 in Papyan et al. (2020), $\mathbf{ZQ}$ constitutes an ETF. As $\mathbf{ZQ}$ directly represents the orthonormal span of $\mathbf{Z}$'s column space, the conclusion follows. □

Gram matrix plays a key role in connecting our metric with Section 6, i.e., understanding the rank-increasing phenomenon.

**Theorem E.4.** *The effective rank of the total sample Gram matrix can be effectively estimated by batch.*

*Proof.* Note scaling does not change effective rank. Change the order of $\mathbf{Z}^\top \mathbf{Z}$ to $\mathbf{ZZ}^\top$, then can rewrite self-correlation as the empirical estimation of expected self-correlation by samples in a batch. This explains the estimation given by Zhuo et al. (2023). □

Interestingly, the following theorem connects our metrics with the Gram matrix.

**Theorem E.5.** *Assuming the dataset is class-balanced and the global mean is $0$, then the effective rank of the covariance matrix of all $K$ class-mean vectors is exactly the same as the effective rank of the Gram matrix.*

*Proof.* As $\mathbf{ZZ}^\top$ and $\mathbf{Z}^\top \mathbf{Z}$ have the same non-zero eigenvalues, thus having the same effective rank. □

# F  DETAILS ON EXPERIMENTS

**Model Architectures.** Similar to MEC (Liu et al., 2022), we select one branch of the Siamese network as the online network and the other branch as the target network, updating the parameters using the exponential moving average method instead of loss backward. We use ResNet50 (He et al., 2015) without the last linear layer as the backbone encoder, whose output feature dimension is 2048. Then we use a three-layer MLP with BN(Batch Normalization) (Ioffe & Szegedy, 2015) and ReLU (Nair & Hinton, 2010) as the projector after the encoder, and the projector maintains the feature dimension to be 2048 through three layers. For the online network, we apply an extra two-layer MLP with BN (Ioffe & Szegedy, 2015) and ReLU (Nair & Hinton, 2010) with hidden dimension 512 and output dimension 2048.

**Optimization and hyperparameters.** For pre-training, we use SGD optimizer with 2048 batch size, $10^{-5}$ weight decay, 0.9 momentum, and 4.0 base learning rate, which is scheduled by cosine decay learning rate scheduler (Loshchilov & Hutter, 2016), to optimize the online network over training process. For the momentum used for the exponential moving average process, it is set to be 0.996 to 1 scheduled by another cosine scheduler. As for linear evaluation, we use LARS optimizer (You et al., 2017) with 4096 batch size, 0.9 momentum, no weight decay, and 0.03 base learning rate scheduled by cosine decay learning rate scheduler, to train the linear layer over 100 epochs, and report the performance of last epoch.

## F.1  ABLATION STUDIES

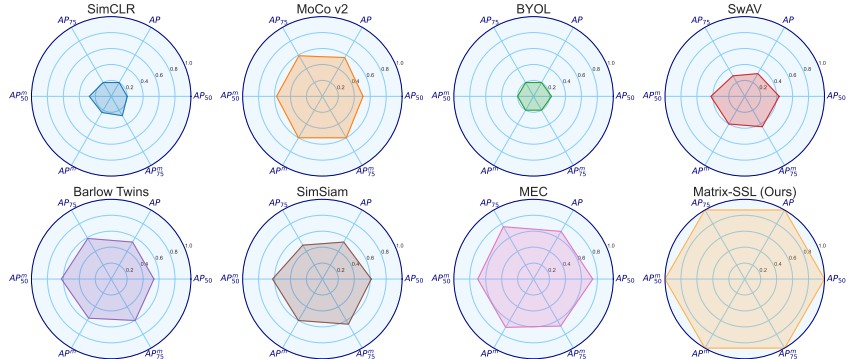

Figure 4: Comparative performance of Matrix-SSL and other state-of-the-art methods on COCO Detection and COCO Instance Segmentation tasks (Lin et al., 2014). Each axis represents a performance metric ($AP$, $AP_{50}$, etc.), and we denote $AP^{\text{mask}}$ by $AP^{\text{m}}$. Matrix-SSL significantly outperforms other methods across all metrics up to 3.3% with only half pre-training epochs, demonstrating its efficacy in transfer learning tasks. For clarity, the maximum value for each metric is scaled to 1, and the minimum is scaled to 0.2, following the same visualization styles from Yu et al. (2022); Wang et al. (2022a); Liu et al. (2022) .

**Alignment loss ratio.** We first investigate the impact of different alignment loss ratios (i.e., the $\gamma$ in Eqn. 22) on performance. We chose the 100-epoch pre-training task for the ablations, and the results are summarized in Table 4.

Table 4: Linear probing accuracy (%) of Matrix-SSL with various $\gamma$.

| $\gamma$ | 0 | 0.3 | 0.5 | 0.6 | 1 | 1.3 | 1.5 |
|---|---|---|---|---|---|---|---|
| Acc. | 70.6 | 70.7 | 71.0 | 70.9 | **71.1** | 70.8 | 70.8 |

Interestingly, setting $\gamma = 1$ exactly achieves the best linear probe performance, so we set the ratio to be 1 as the default.

**Taylor expansion order.** We investigated the effect of the Taylor expansion order of matrix logarithm implementation on linear evaluation tasks. We keep most of the settings in 7 unchanged, except the Taylor expansion order. The results are summarized in Table 5. As shown in the table, we found that Matrix-SSL performs best when the Taylor expansion order is 4, so we chose 4 as the default parameter.

Table 5: Results of different Taylor expansion orders for Linear evaluation results.

| Taylor expansion order | 3 | 4 | 5 |
|---|---|---|---|
| Top-1 accuracy | 70.9 | **71.1** | 71.0 |

### F.2 EXPERIMENTS OF DIMENSIONAL COLLAPSE

We measure dimensional collapse on various self-supervised learning methods, including SimCLR (Chen et al., 2020a), BYOL (Grill et al., 2020), Barlow Twins (Zbontar et al., 2021) and SimSiam (Chen & He, 2021) with or without stop gradient. We reproduce the above methods on the self-supervised learning task of CIFAR100 (Krizhevsky et al., 2009) dataset, using the open source implementations (Tsai et al., 2021a; Hua, 2021) of the above methods tuned for CIFAR. After pre-training, we use the saved checkpoints to evaluate the results of these methods on different metrics.

We calculate the intra-class and inter-class effective rank directly by definition, while for MCE, we shuffle the testing dataset, import the data with 512 batch size, and finally output the average metrics of all batches.

We perform t-SNE (van der Maaten & Hinton, 2008) visualization on the last checkpoint of each method with the help of scikit-learn (Pedregosa et al., 2011). We use the default t-SNE (van der

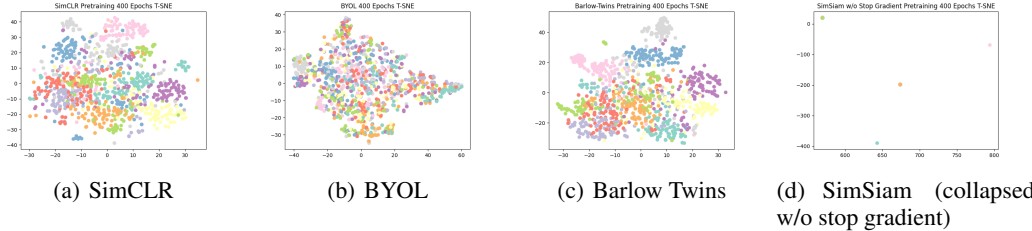

(a) SimCLR     (b) BYOL     (c) Barlow Twins     (d) SimSiam (collapsed w/o stop gradient)

Figure 5: Visualization of feature representation for images in 10 different classes from CIFAR-100 dataset via t-SNE of various self-supervised learning methods. We find that in many categories, it is difficult to distinguish between two non-contrastive methods (BYOL, Barlow Twins) and contrastive method (SimCLR) by t-SNE.

Maaten & Hinton, 2008) parameter of scikit-learn (Pedregosa et al., 2011) and select the first 5 or 10 categories from 100 categories in CIFAR-100 (Krizhevsky et al., 2009) for visualization.

