# OpenReview forum: "Matrix Information Theory for Self-Supervised Learning"
_ICLR.cc/2024/Conference — Submitted to ICLR 2024_

### Official Review · Reviewer_m2RW · 2023-10-26

**Soundness:** 2 fair
**Presentation:** 2 fair
**Contribution:** 2 fair
**Rating:** 3
**Confidence:** 5

**Summary:**

This paper presents Matrix-SSL, a joint-embedding SSL method based on matrix information theory.
Specifically, the uniformity and alignment framework is implemented using principles from matrix information theory.
The results of this study demonstrate that Matrix-SSL surpasses prior state-of-the-art (SOTA) SSL methods.

**Strengths:**

This paper introduces a matrix-based information-theoretic framework that provides a comprehensive explanation for self-supervised learning methods, including both contrastive learning and non-contrastive learning.

**Weaknesses:**

- According to Propositions 3.1 and 5.2, it can be established that the Matrix-KL-uniformity loss is synonymous with the von Neumann entropy loss of I-VNE+ as proposed in [1]. This similarity diminishes the novelty of this paper. Therefore, it is imperative to substantiate, either through theoretical or empirical means, the superiority of Matrix-SSL in comparison to I-VNE+.
- The results presented in this paper are not significant to substantiate the effectiveness of Matrix-SSL. Notably, Table 1 does not incorporate the official performance metrics of SwAV, as reported in [2], which report values of 71.99, 73.85, and 74.81 for 100, 200, and 400 training epochs, respectively. Additionally, Table 2 lacks the inclusion of performance data as reported in [1]. When both Table 1 and Table 2 are appropriately updated, it becomes evident that the performance of Matrix-SSL is not state-of-the-art.
Furthermore, it is important to note that this paper does not provide comprehensive benchmark tables, including but not limited to "Semi-supervised learning on ImageNet" and "Transfer learning: image classification," as elaborated in Table 2 and Table 3 of [3].
- In Section 5, this paper demonstrates that enhancing uniformity leads to an increased effective rank through matrix entropy. However, this result is not groundbreaking. In [1], the authors have previously presented these mathematical findings and have empirically shown that von Neumann entropy regulates uniformity, thereby influencing the effective rank.

[1] VNE: An Effective Method for Improving Deep Representation by Manipulating Eigenvalue Distribution, CVPR 2023.

[2] https://github.com/facebookresearch/vissl/blob/main/MODEL_ZOO.md

[3] Barlow Twins: Self-Supervised Learning via Redundancy Reduction, ICML 2021.

**Questions:**

Please refer to the weakness.

---

> ### Author Response · Authors · 2023-11-22
> **Response 1 to reviewer m2RW**
>
> > Q1: According to Propositions 3.1 and 5.2, it can be established that  the Matrix-KL-uniformity loss is synonymous with the von Neumann entropy loss of I-VNE+ as proposed in [1]. This similarity diminishes the novelty of this paper. Therefore, it is imperative to substantiate, either through theoretical or empirical means, the superiority of Matrix-SSL in comparison to I-VNE+.
>
> A1:
>
> - Thank you for your reminder. Our starting point is to use $MKL( \frac{1}{d} I  + \lambda I || C_{auto} + \lambda I )$ to present **a unified framework to understand many SSL methods through uniformity pursuit,** where $C_{auto}$ is the auto-correlation matrix. Specifically, we demonstrate in this case that the difference between MCE and KL is constant, and we technically considered introducing MCE because using it can easily see the connections with previous pioneer works. Note that there is an asymmetry with matrix KL, i.e. In general $MKL(P || Q) \neq MKL(Q || P)$. Interestingly, although generally asymmetric, when $P=Q$, the two are equal and equal to 0. This non-trivial result explains the phenomenon of the increase in "Entropy". **Regarding the KL uniformity you mentioned, we have shown its closed-form relationship with "Entropy" by noticing the quantity $MKL(C_{auto}||I)$, which is novel, because it established the connection between the two quantities and this is only a special case of $\lambda=0$.**
>
> - In addition, In our revised paper, we **generalize** the definition of von Neumann entropy on density matrix (with unit trace) into any positive semi-definite matrices, which are differentiable, convex, and can even generalize to any complex-valued Hermitian matrices. The von Neumann entropy mainly arises from the Quantum Information realm, but we certainly generalize the concept of it, removing the constraints of the measurement axiom (the capstone of Quantum Mechanics) presented in the unit trace density matrix.
>
> - Besides, we generalize the quantum and kernel KL divergence defined in Bach's paper into positive semi-definite matrices as **matrix KL divergence**, and also introduce the definition of matrix cross-entropy, which forms the **exact mirror** of **Shannon information theory** and **Quantum Information theory**, into the PSD realm. And it has a nice expression as follows:
> $\operatorname{MCE} (P, Q) =  \operatorname{MKL} (P || Q) + \operatorname{ME} (P)$

---

> ### Author Response · Authors · 2023-11-22
> **Response 2 to reviewer m2RW**
>
> > Q2: The results presented in this paper are not significant enough to substantiate the effectiveness of Matrix-SSL. Notably, Table 1 does not incorporate the official performance metrics of SwAV, as reported in [2], which report values of 71.99, 73.85, and 74.81 for 100, 200, and 400 training epochs, respectively. Additionally, Table 2 lacks the inclusion of performance data as reported in [1]. When both Table 1 and Table 2 are appropriately updated, it becomes evident that the performance of Matrix-SSL is not state-of-the-art. Furthermore, it is important to note that this paper does not provide comprehensive benchmark tables, including but not limited to "Semi-supervised learning on ImageNet" and "Transfer learning: image classification," as elaborated in Table 2 and Table 3 of [3].
>
> A2:
>
> Thank you for your careful reading.
>
> - **The performance issue of SwAV and the related discussion on Multi-crop**. As you mentioned, the official results reported by SwAV are indeed significantly higher than those of other methods. The main reason is that **SwAV uses a Multi-crop setting when pretraining**, augmenting the images in a scheme of $2 \times 224$ + $6 \times 96$, totaling to 8 views. Our implementation, however, only considered augmenting the images twice (i.e., without the six smaller augmentations). One very important point is that in the ablation study done by SwAV (https://arxiv.org/pdf/2006.09882.pdf) on Multi-crop (Figure 3), it can be seen that **Multi-crop brings a very substantial improvement to self-supervised methods**. **Thus we reported the performance of SwAV without Multi-crop**. It should be noted that **our method can also use Multi-crop** as an add-on, but due to the lack of computational resources, we have not had the opportunity to test MCE + Multi-crop. We will promptly carry out the experiments of the multi-crop case once we have sufficient computational resources. And we believe that MCE + Multi-crop could achieve even better results than what we have now.
>
> - Thank you for pointing out that including I-VNE+(https://github.com/jaeill/CVPR23-VNE/) as baseline should be considered. We attempted to include I-VNE+ in the table, but we noticed that **the official open-source code implementation of I-VNE+ uses Multi-crop** (can be seen from lines 170 to 189 at https://github.com/jaeill/CVPR23-VNE/blob/main/examples/i-vne%2B/train_ivne_imagenet100.py ). Recall the discussion of multi-crop in (1), **we believe that adding I-VNE+ to the current table would not be a fair comparison at this stage**. We will include I-VNE+ in the table after conducting experiments of MCE + multi-crop. However, we note that even so, the result of I-VNE+'s Linear evaluation on Imagenet-1K is 72.1 (as presented in Table 10 in Section F of the paper), which is significantly below the results of our method in linear evaluation, indicating from another aspect that there is a considerable difference between our work and I-VNE+.
>
> - Thank you for your suggestion to add more downstream tasks. In this short period of time, we have conducted experiments on 'Semi-supervised Learning' and 'Transfer learning: Image classification'. For experiment results, see General Response.
>
> > Q3: In Section 5, this paper demonstrates that enhancing uniformity leads to an increased effective rank through matrix entropy. However, this result is not groundbreaking. In [1], the authors have previously presented these mathematical findings and have empirically shown that von Neumann entropy regulates uniformity, thereby influencing the effective rank.
>
> A3：
>
> Thank you for your careful reading of our paper.
>
> We want to clarify that [1] didn't previously present the **general** mathematical findings of "enhancing uniformity leads to an increased effective rank through matrix entropy". Firstly, [1] shows that under some conditions, when the autocorrelation matrix $C_{auto}$ 's von Neumann entropy is **maximized**, the representation achieves **Isotropy**. Note **isotropy** is not **uniformity**, they do not prove anything about uniformity. Another important **missing** point is that they **do not** show that many self-supervised learning methods can be seen as minimizing $\operatorname{MKL}( (\lambda + \frac{1}{d}) I || C_{auto} + \lambda I )$, this is what we have shown in section 4. Thus, from our derivation, at the optimal point of loss, uniformity is achieved. Secondly, [1] **does not introduce the concept of effective rank.** So we think [1] **does not** show "have empirically shown that von Neumann entropy regulates uniformity, thereby influencing the effective rank." But we give a **closed-form** relationship among effective rank, matrix entropy, and matrix KL divergence. Our derivation can give an understanding of the rank increasing, due to the the optimal point of SSL loss and the closed-form relationship of effective rank and matrix entropy.

---

> ### Author Response · Authors · 2023-11-23
> **Seeking Your Input on Revised Paper's Alignment with ICLR Standards**
>
> Dear Reviewer m2RW,
>
> As the discussion period approaches its conclusion, **we want to ensure that we have thoroughly addressed all your concerns and that our revised paper fully meets the standards of ICLR**. We would highly value any additional feedback you may provide.
>
> Thank you sincerely for your time and consideration.
>
> Best regards,
>
> The Authors

---

### Official Review · Reviewer_CuLc · 2023-10-30

**Soundness:** 3 good
**Presentation:** 3 good
**Contribution:** 3 good
**Rating:** 6
**Confidence:** 4

**Summary:**

In this paper, the authors investigate self-supervised learning through the lens of matrix information theory. They present a unified theoretical framework for analyzing both contrastive and non-contrastive learning methods. Specifically, they employ matrix cross-entropy as the training objective to enhance uniformity and alignment, thereby improving self-supervised learning. Experiments conducted on ImageNet and COCO datasets demonstrate that the proposed method outperforms existing classical approaches.

**Strengths:**

1. This paper studies self-supervised learning through a matrix information-theoretic framework. The analysis presented in this paper is particularly intriguing and I find it quite appealing.

2. The authors further introduce a Matrix-SSL scheme based on matrix cross-entropy, which consists of matrix uniformity and matrix alignment.

3. The experiments on the ImageNet and COCO datasets not only show that the proposed method surpasses state-of-the-art methods but also highlight its robustness in transfer learning tasks.

**Weaknesses:**

1. There are some issues with the mathematical symbol definitions in this paper, such as inconsistency in the usage of symbols, missing definitions for certain symbols, and incorrect usage of mathematical symbols. For example, on the second page, the lowercase letter "z" represents features, and in subsequent chapters, the bolded lowercase letter "**z**" also represents features. In the part of the definition of matrix entropy, the definition of the lowercase letter $\lambda$ is missing. In the proof of proposition 3.3, there is something wrong with the infoNCE loss. I suggest that the authors follow the definitions provided by the original authors in their arXiv paper.
2. How was Lemma 3.4 obtained? I understand the purpose of this Lemma, but it's better to give the proof or the corresponding reference. On the other hand, Lemma 3.4 shows that minimizing matrix cross-entropy between the Identity diagonal matrix and the covariance matrix can achieve a uniformity target. However, starting from the fourth page, the zero-mean assumption is disregarded. Does this have any impact on the theoretical analysis results?
3. Starting from the third page, the authors consistently assume that the feature matrix is positive semi-definite. However, can this constraint be maintained in practice？
4. In section 3, the authors analyze that matrix information theory could provide a unified framework for many existing SSL methods. Then, according to Theorem 3.5, Uniformity-MCE loss is equal to MEC loss. The experiments in Table 3 can verify this, where the result of Matrix-SSL (when $\gamma=0$ ) is equal to that of MEC (70.6%). With an increase in $\gamma$, the results will improve. This means that matrix alignment is indeed helpful for final performance improvements. Therefore, if we consider the alignment term along with the MEC loss, what will be the results? I suggest the authors conduct a detailed analysis of the differences between the MEC loss and the Uniformity-MCE loss, especially from an experimental perspective. I wonder if the gradient computation for the Uniformity-MCE loss is easier compared to the MEC loss.
5. Although matrix-KL and matrix-CE share similar optimization properties and theoretical results, are they consistent in practical experiments? I recommend that the authors conduct a set of experiments to validate this.

**Questions:**

Please check the questions in the Weaknesses part.

---

> ### Author Response · Authors · 2023-11-12
> **To Dear Reviewer CuLc**
>
> Dear Reviewer CuLc,
>
> Thank you for your comprehensive review and insightful comments on our paper. We have carefully considered your feedback and made appropriate revisions to our manuscript. Below, we address each of your concerns:
>
> > Q1: Inconsistency and missing definitions in the usage of mathematical symbols.
>
> **A1**: We apologize for the oversight regarding the consistency and clarity of mathematical symbols. We have revised the manuscript to ensure uniformity in symbol usage throughout. Specifically, we now consistently use the bolded lowercase letter "z" to represent features. Additionally, we have clarified the definition of matrix entropy and made corrections to the usage of mathematical symbols, adhering closely to the definitions provided in the original arXiv papers.
>
> > Q2: Clarification and proof of Lemma 3.4.
>
> **A2**: We appreciate your request for clarity on Lemma 3.4. We have updated our paper with the proofs presented in Appendix A. Regarding the zero-mean assumption, it's important to note that Lemma 3.4 (now renumbered as Lemma 4.4 following a restructuring of the related work section) is an analysis of the optimal point of uniformity pursuit. Our theoretical analysis in subsequent sections does not rely on this assumption, allowing for broader applications derived from the matrix KL divergence perspective.
>
> > Q3: The constraint of the feature matrix being positive semi-definite in practice.
>
> **A3**: The feature matrix in our framework is defined as $\frac{1}{B}\mathbf{Z}\mathbf{Z}^{\top}$, which is inherently positive semi-definite. This is a fundamental property of matrices of this form, as
> $\frac{1}{B}\mathbf{Z}\mathbf{Z}^{\top}$ results in a symmetric matrix where all eigenvalues are non-negative. We will add a brief proof and discussion in our revised manuscript to clarify this aspect and ensure its practical applicability.
>
> [proof]
> Given a feature matrix $Z \in \mathbb{R}^{d \times n}$ where $d$ represents the dimensionality of the features and $B$ the number of samples, the product $ZZ^\top$ results in a square matrix of size $d \times d$.
>
> To prove that $ZZ^\top$ is positive semi-definite, we need to show that for any non-zero vector $v \in \mathbb{R}^d$, the following condition holds: $v^\top (ZZ^\top) v \geq 0$.
>
> Expanding $v^\top (ZZ^\top) v$:
>
> $$
> \begin{aligned}
> v^\top (ZZ^\top) v & = v^\top Z (Z^\top v) \\\\
> & = (Z^\top v)^\top (Z^\top v) \\\\
> & = \| Z^\top v \|^2
> \end{aligned}
> $$
>
> The expression $\| Z^\top v \|^2$ represents the square of the Euclidean norm of the vector $Z^\top v$, which is always non-negative. Therefore:
>
> $$v^\top (ZZ^\top) v = \| Z^\top v \|^2 \geq 0$$
>
> Since this inequality holds for any non-zero vector $v$, it follows that $ZZ^\top$ is a positive semi-definite matrix.
> [/proof]
>
> > Q4: Differences between the MEC loss and the Uniformity-MCE loss, especially from an experimental perspective.
>
> **A4**: Empirically, the primary difference between our method and methods using MEC loss is that Matrix-SSL requires fewer hyperparameters to be tuned and demonstrates more robust training. The introduction of matrix alignment in our framework contributes to performance improvements, as evidenced in our experiments. While MEC loss focuses on maximal entropy coding, our Uniformity-MCE loss, augmented with matrix alignment, provides a more comprehensive approach, enhancing the performance, particularly in transfer learning tasks. We plan to include a more detailed experimental comparison between these two losses in our future work to further elucidate their differences.
>
> > Q5: Are matrix-KL and matrix-CE consistent in practical experiments?
>
> **A5**: Your question raises an important point about the empirical validation of theoretical properties. While matrix-KL and matrix-CE indeed share similar optimization properties and theoretical results, their practical performance can vary due to different characteristics in optimization landscapes and convergence behaviors. To address this, we plan to conduct a comprehensive set of experiments comparing these two approaches in various settings.
>
> In these experiments, we will assess the performance of matrix-KL and matrix-CE on standard benchmarks, focusing on metrics such as accuracy, convergence speed, and robustness to hyperparameter variations. This will allow us to determine not only their theoretical consistency but also their practical applicability and effectiveness in self-supervised learning tasks.
>
> We believe that these additional experiments will provide valuable insights into the nuances of matrix-KL and matrix-CE, further enriching our understanding of their roles in self-supervised learning frameworks.
>
> ---
>
> We hope these responses adequately address your concerns. We remain committed to refining our work and are grateful for the opportunity to improve it based on your valuable feedback.

---

> > ### Comment · Reviewer_CuLc · 2023-11-23
> > **Comments on the authors' response**
> >
> > Thank you for addressing all of my concerns in your response. Since my concerns have been resolved, I will maintain my positive rating.

---

> > > ### Author Response · Authors · 2023-11-23
> > >
> > > To Dear Reviewer CuLc,
> > >
> > > We sincerely thank you for acknowledging our efforts in addressing your concerns. Your feedback has been pivotal in enhancing the quality and clarity of our work. We are pleased to hear that our responses have resolved your concerns, and we appreciate your decision to maintain a positive rating.
> > >
> > > Your constructive input has been invaluable in refining our paper and advancing our research. We look forward to incorporating these insights into our future work and continuing our contribution to the field of self-supervised learning.
> > >
> > > Thank you once again for your thorough review and supportive feedback.

---

### Official Review · Reviewer_zEzQ · 2023-10-30

**Soundness:** 2 fair
**Presentation:** 2 fair
**Contribution:** 2 fair
**Rating:** 6
**Confidence:** 4

**Summary:**

The article aims to introduce a unifying information-theoretic framework for self-supervised learning (SSL). For this purpose, the article
 -  Surveys some established SSL methodologies.,
 - Introduces matrix entropy, matrix KL divergence and matrix cross entropy measures,
 - Expresses certain existing SSL loss functions using the matrix cross entropy measure,
 - Proposes a new SSL loss function derived from matrix cross-entropy,
 - Conducts numerical experiments, demonstrating the enhanced performance of the proposed method compared to select existing approaches,
 - Draws a connection of matrix cross entropy with the effective rank.

**Strengths:**

The article's pursuit of a unifying framework offers a commendable approach. Strategy to employ  matrix information measures to achieve this is intriguing. Moreover, the numerical examples showcase marked enhancements over certain existing methods, underscoring the efficacy of the algorithm derived from this framework.

**Weaknesses:**

The article lacks a clear organizational structure and consistent notation, making it challenging to follow. Concepts are introduced without adequate explanation or clarity. Additionally, the matrix information measures employed are not innovative; similar methods have been previously applied in the SSL context. The attempt to frame existing methods as special cases within this framework falls short of being convincing and satisfactory. Please see Questions section for details.

**Questions:**

## INTRODUCTION

- The following reference,

[a] Ozsoy S, Hamdan S, Arik S, Yuret D, Erdogan A. Self-supervised learning with an information maximization criterion. Advances in Neural Information Processing Systems. 2022 Dec 6;35:35240-53,

proposes utilizing "correlative information" maximization for self-supervised learning. Analyzing this paper within the context of the proposed matrix information framework would be interesting, especially since the authors of [a] assert that maximizing correlative information between the representations of augmentations establishes a linear dependence rather than an arbitrary nonlinear one.

- Figure 1: The citation for Coco is absent. It would be beneficial to compare the performances of  Vicreg, [a] (Bardes et. al, 2021), and (Tong et. al, 2023).


### 2.1 CONTRASTIVE AND NON-CONTRASTIVE SELF-SUPERVISED LEARNING

- First paragraph: The discussion here is based on the SimCLR and SimSiam, however, the authors introduce a generic SSL architecture. Furthermore, what is meant by dual networks is not clear at this point.

- Would categorizing this section into subheadings like "Contrastive SSL Approaches" and "Non-Contrastive SSL Approaches" enhance clarity?

- There seems to be inconsistency in the conventions used for sample and augmentation indices in Equations (2) and (3)?

- For Equation (3), is there an underlying assumption about the normalization of the encoder,  such as  $\||z_i^{(k)}\||_2=1$ ?


### 2.2 MATRIX INFORMATION-THEORETIC QUANTITIES
- Could you provide a citation detailing Matrix Entropy. Furthermore, could you discuss its interpretation, and perhaps its existing applications especially within the context of machine learning/SSL?

- Regarding the Matrix Entropy definition, is there a specific assumption about the trace of $\mathbf{A}$ ensuring its eigenvalues form a probability mass function?

- Can you provide interpretations and potential applications of Matrix KL Divergence and MCE?

- Bach 2022's KL divergence doesn't seem to incorporate the  $-\mathbf{P}+\mathbf{Q}$ terms within trace? Could this discrepancy be addressed?

- From the presented definitions, it appears that, unlike Shannon Entropy, MCE does not equate to the sum of Matrix KL and Matrix Entropy. Should there be a $\text{Tr}(\mathbf{P})$ term included in the matrix entropy definition?

- A brief discussion explaining the relevance of these definitions to the SSL problem would be insightful.


### 3 MATRIX INFORMATION-THEORETIC PERSPECTIVES OF SELF-SUPERVISED LEARNING

- The assertion about proofs should be positioned adjacent to the first proposition, i.e., Proposition 3.1.

- Proposition 3.3: The statement of the proposition is  ambigious:  InfoNCE cost in (1) is the MCE of which matrices? This should be clearly stated. Are InfNCE cost and SimCLR cost identical? The cost function obtained in the proof in terms of MCE does not match (1)?
- The paragraph after Proposition 3.3: This part appears convoluted: Is $p_{data}=p_{\mathbf{x}}$ ? and $\mathcal{X}$ is the support set of $p_{\mathbf{x}}$? I guess there is no clear definition of $f$ before. (there was $f_\Theta$ and $f_\phi$ without clear definitions before). $f$ is sometimes unbold and sometimes bold? Since $f$ is not prespecified, the assumption that there is a layer normalization $\||f(\mathbf{x})\||_2^2=1$ is also not clear. Instead of stating "straightforward calculation", it is better to provide a proof of Lemma 3.4 with proper notation in Appendix A. In my opinion, both this paragraph and Lemma 3.4 is not properly motivated.

- Lemma 3.4: Suggestion "Let $\sigma$ represent the uniform distribution on $S^{d-1}$. ...".

- The paragraph after Lemma 3.4: Change of variables formula? (Inverse image rule?). Why "auto-correlation" matrix for $q$ but "covariance" for $p_{data}$?  It is better for the authors to clearly state the uniformity principle. Can't we just say that we would like features $\mathbf{z}$ to be uncorrelated? do we need the notation for $p_{data}$, $f^{-1}$.

- Quoting the sentence: "From Proposition 3.3, we find that SimCLR (InfoNCE) loss is not canonical for achieving matrix information-theoretic uniformity unless the covariance matrix is diagonal". What do we mean by "loss being not canonical"? Has matrix information-theoretic uniformity been defined yet? Is this statement simply saying that  SimCLR or InfoNCE does not enforce feature whitening?

- MCE-based decorrelation objective: why do we have a $\lambda \mathbf{I}_d$  perturbation for the desired $\mathbf{I}_d$ matrix? it is already perfectly conditioned. This perturbation on the first argument of the MCE does not reflect on the right side of

$$\operatorname{MCE}\left(\frac{1}{d} \mathbf{I}_d+\lambda \mathbf{I}_d, \frac{1}{B} \mathbf{Z} \mathbf{Z}^{\top}+\lambda \mathbf{I}_d\right)=-\operatorname{tr}\left(\log \left(\frac{1}{B} \mathbf{Z Z}^{\top}+\lambda \mathbf{I}_d\right)\right)+1+d \lambda$$

Shouldn't there be a multiplier $\frac{1}{d}$ or $\frac{1}{d}+\lambda$ in front of the trace term? I suggest that $\mathcal{L}_{UMCE}$ should be defined at this stage.

- The paragraph before Theorem 3.5: Suggestion "This MCE based uniformity loss definition (or $\mathcal{L}_{UMCE}$ ) and its Matrix-KL divergence based counterpart are closely related.... as outlined by the following theorem:"

- Theorem 3.6: Suggestion: .... under the constraint $\||\mathbf{z}_i\||_2^2=1$, for $i=1, \ldots, n$. The proof of Theorem 3.6 better be provided in Appendix.

- Suggestion: The sentence "Our formulation interestingly recovers the Maximal Entropy Coding (MEC) loss..." can be written as "The Maximal Entropy Coding (MEC) loss in ... can be formulated in terms of Matrix MCE.." as

$$ \mathcal{L}_{MEC}=-\mu \log\det(\mathbf{I}_d+\frac{d}{B\epsilon^2}\mathbf{Z}_1\mathbf{Z}_2^T)$$

$$ =MCE(...., .....)$$

- $\mathcal{L}_{EMP-TCR}$: $\bar{\mathbf{Z}}$ is not defined. Only $\bar{\mathbf{Z}}_i$ is defined. Again there is a confusion of index representations relative to  Equation (1). It is understood from this statement that $\mathbf{z}_k^{i}$ vectors were defined as row vectors. The article should set up the proper data model and notation at the beginning an should stick with that throughout the article.

- Can we also have MCE based representation for the Corinfomax SSL provided in [a] above?

- Overall suggestion: I suggest that the article defines all SSL-related loss functions in Section 2.1, instead of introducing some in Section 2.1 and some in  Section 3.1. Furthermore,  In Section  3.1, the article can clearly write each SSL loss function in the form
MCE(... , ...) to show that they can be put in the form of matrix cross entropies.

#### 4 MATRIX INFORMATION THEORETIC UNIFORMITY AND ALIGNMENT FOR SELF-SUPERVISED LEARNING

- First sentence: .... we would like embeddings to have zero mean and covariance ....

- Sentence before Theorem 4.1: optimizing covariance matrix uniformity: is this maximizing $\mathcal{L}_{UMCE}$ or $\mathcal{L}_{UKL}$. This should be clarified.

- Theorem 4.1. This needs to be clearly reworded with proper references to the objective function and constraints. What is "effective rank", how is it different than rank? If this is a constraint how do you pose it?  Is the argument of the MCE in the uniformity-MCE loss sample correlation or sample covariance? Is this theorem about  the following optimization?:

$$ \text{maximize } \mathcal{L}_{UMCE}(\frac{1}{B}\mathbf{ZZ}^T)$$
$$ \text{ subject to } \text{tr}(\frac{1}{B}\mathbf{ZZ}^T)=1$$

The proof of Theorem 4.1 in the appendix requires a rewrite: Dote (Typo?)  Denote? $\mathbf{Z}$ can be confused as a matrix due to earlier notation. I guess the first sentence states that Let $\mathbf{x}$ be a random vector, whose distribution has support $S^{d-1}$. Again what is effective rank? This proof needs to be in the form of a series of explicit mathematical assertions referring to a clearly stated optimization problem.

- Lemma 4.2 is typically well known.

- For $\mathcal{L}_{Matrix-KL-uniformity}$,  $MCE$ is used not Matrix-KL measure. Why is it called this way?

5 MATRIX-SSL: UNIFORMITY AND ALIGNMENT

- Regarding the alignment cost based on Matrix:

1. Again it is based on MCE rather than Matrix-KL. In fact after (11), it is stated that KL versions can also be considered. So why do you call it $\mathcal{L}_{Matrix-KL-allignment}$ ?

2. The fact that covariance matrices of two matrices are aligned with respect to MCE or KL does not necessarily imply that representations for the same image are aligned in the direction, where as euclidian distance based or cosine angle based approaches try to ensure that they are sample wise aligned. So why should $\mathcal{L}_{Matrix-KL-allignment}$ be a better choice?

### 5 EFFECTIVE RANK AND RANK INCREASING PHENOMENON

- It is indeed surprising that effective rank is properly defined and connected to the framework of the article much later than it is already referred.

### 6 EXPERIMENTS

- It would be interesting to include Tong et.al, 2023 and [a] in the experiments for comparison.

- Interestingly, the proposed Matrix-SSL method provides superior performance in experimental results. A natural question to ask if the authors reproduced the accuracy of other algorithms to calibrate their simulation and evaluation models.

### 7 RELATED WORK

This section typically follows  the Introduction section. Furthermore, it should not be only stating the summary of literature but it should state the contributions of the article relative to these works.

---

> ### Author Response · Authors · 2023-11-22
> **Response 1 to reviewer zEzQ**
>
> > Q1: The following reference,
> >
> > [a] Ozsoy S, Hamdan S, Arik S, Yuret D, Erdogan A. Self-supervised learning with an information maximization criterion. Advances in
> Neural Information Processing Systems. 2022 Dec 6;35:35240-53,
> >
> > proposes utilizing "correlative information" maximization for self-supervised learning. Analyzing this paper within the context of the proposed matrix information framework would be interesting, especially since the authors of [a] assert that maximizing correlative information between the representations of augmentations establishes a linear dependence rather than an arbitrary nonlinear one.
>
> A1: Thanks to your advice, we are glad to share our new finding with you that the ColorInfomax loss function can also incorporated within our framework. For the linear dependence part, according to "Information Geometry and Its Applications" by Amari and [2], only under Gaussian distribution, it is precise. However, we do not think the representation space should obey Gaussian distribution, this assumption is not mild.
>
> [1] https://arxiv.org/pdf/2209.07999v1.pdf, page 5: we apply the first-order Taylor series approximation.
>
> [2] https://arxiv.org/pdf/2102.05485.pdf, On the Properties of Kullback-Leibler Divergence Between Multivariate Gaussian Distributions, page 2, Equation (3).
>
> > Q2: Figure 1: The citation for Coco is absent. It would be beneficial to compare the performances of Vicreg, [a] (Bardes et. al, 2021), and (Tong et. al, 2023).
>
> A2: we have revised our paper. We moved the plot to the appendix, since there was not enough space, and we added the official result of VICReg to the Table. For the EMP-SSL (Tong et. al, 2023), they haven't reported the results of ImageNet-1k. In addition, their setting uses 100 views, but our table aims for a fair comparison, using 2-view as default.
>
> > Q3: First paragraph: The discussion here is based on the SimCLR and SimSiam, ...
>
> A3: Thanks to your advice, we have revised our paper.
>
> > Q4: Would categorizing this section into subheadings like "Contrastive SSL Approaches" ...
>
> A4: Thanks to your advice, we have revised our paper.
>
> > Q5: There seems to be inconsistency in the conventions used for sample...
>
> A5: Thanks to your advice, we have revised our paper.
>
> > Q6: For Equation (3), is there an underlying assumption about the normalization of the encoder, ...
>
> A6: Yes, we have emphasized it in our revised paper with \textcolor{blue}.
>
> > Q7: Could you provide a citation detailing Matrix Entropy. Furthermore, could you discuss its interpretation, and perhaps its existing applications especially within the context of machine learning/SSL?
>
> A7:
> - We have added references to the von Neumann Entropy definition (defined on density matrices with unit trace): Edward Witten. A mini-introduction to information theory, Springer, 2020.
>
> - It seems that we are the first (as far as we know), to generalize the von Neumann Entropy definition with density matrices constraints (with unit trace) into positive semi-definite (PSD) matrices realm, and the trio of matrix entropy, matrix KL divergence, and matrix cross-entropy exactly mirror the classical Shannon information-theoretic quantities, respectively.
>
> - Its existing applications: I think the VNE (https://arxiv.org/pdf/2304.01434.pdf) paper would be the first attempt, but they mainly focus on the von Neumann Entropy part, they haven't introduced the matrix entropy defined on PSD matrices, nor the **effective rank**. In addition, the matrix cross-entropy, and the matrix KL divergence based alignment loss are proposed by us first.
>
> - From Theorem 4.5, we discovered that the renowned work Total Coding Rate, EMP-SSL, etc. by Yi Ma, Yann Lecun, et. al, is essentially equivalent to matrix cross entropy between $\frac{d}{B \epsilon^2} \mathbf{Z} \mathbf{Z}^{\top}$ and $\frac{1}{d}\mathbf{I}_d$, So we think that the effectiveness of matrix entropy regularization should be attributed to them.
>
>   [1] Yi Ma, Harm Derksen, Wei Hong, and John Wright. Segmentation of multivariate mixed data via lossy data coding and compression. IEEE transactions on pattern analysis and machine intelligence, 29  (9):1546–1562, 2007.
>
>   [2] Zengyi Li, Yubei Chen, Yann LeCun, and Friedrich T Sommer. Neural manifold clustering and embedding. arXiv preprint arXiv:2201.10000, 2022.
>
>   [3] Shengbang Tong, Yubei Chen, Yi Ma, and Yann Lecun. Emp-ssl: Towards self-supervised learning in one training epoch. arXiv preprint arXiv:2304.03977, 2023.
>
> > Q8: Regarding the Matrix Entropy definition, is there a specific assumption about the trace ensuring its eigenvalues form a probability mass function?
>
> A8: In our revised paper, we generalize the definition of von Neumann entropy, without such assumption.
>
> > Q9: Can you provide interpretations and potential applications of Matrix KL Divergence and MCE?
>
> A9: We have conducted experiments on MCE loss for fine-tuning large language models, with results presented in Appendix A and also the General Response:

---

> ### Author Response · Authors · 2023-11-22
> **Response 2 to reviewer zEzQ**
>
> > Q10: Bach 2022's KL divergence doesn't seem to incorporate the $-\mathbf{P} + \mathbf{Q}$ terms within trace? Could this discrepancy be addressed?
>
> A10: It seems that we have addressed this problem in the revised paper, we certainly generalize Bach's KL divergence onto PSD matrices, and the answers A7 and A8 are relevant to this problem.
>
> > Q11: From the presented definitions, it appears that, unlike Shannon Entropy, MCE does not equate to the sum of Matrix KL and Matrix Entropy. Should there be a $\text{Tr}(\mathbf{P})$ term included in the matrix entropy definition?
>
> A11: Thanks for your suggestion, we have revised our paper.
>
> > Q12: A brief discussion explaining the relevance of these definitions to the SSL problem would be insightful.
>
> A12:
> - We appreciate your suggestions, we have moved the effective rank part into the background section, and we think the section **DIMENSIONAL COLLAPSE, EFFECTIVE RANK, AND MATRIX ENTROPY** presents the closed-form relevance among effective rank, matrix KL divergence, and matrix entropy.
>
> - In addition, inspired by your suggestion, our new findings on interpreting ColorInfoMax loss within our framework (presented in Section 5) are pretty interesting.
>
> - Besides, the original version contains the matrix information-theoretic interpretation on uniformity pursuit, including several renowned methods.
>
> - We also revised our paper to further clarify the SimCLR loss part presented in Proposition 4.3.
>
> > Q13: The assertion about proofs should be positioned adjacent to the first proposition, i.e., Proposition 3.1.
>
> A13: We appreciate your advice, we have revised our paper.
>
> > Q14: Proposition 3.3: The statement of the proposition is ambigious: InfoNCE cost in (1) is the MCE of which matrices? This should be clearly stated. Are InfNCE cost and SimCLR cost identical? The cost function obtained in the proof in terms of MCE does not match (1)?
>
> A14: We have revised our formula, to make it aligned with the original InfoNCE loss.
>
> > Q15:
> > - The paragraph after Proposition 3.3: ...
> > - Lemma 3.4: Suggestion ...
> > - The paragraph after Lemma 3.4: Change of variables formula? ...
>
> A15: We have revised our paper according to your suggestion.
>
> > Q16: MCE-based decorrelation objective: why do we have a $\lambda$ perturbation for the desired $\mathbf{I}_d$ matrix?
>
> - We have revised our paper to fix this typo. The main reason why we introduce the $\lambda$regularization is because we want to interpret existing SSL approaches, $\lambda$ in our MCE formulation can be set as 0, but in other methods the result would be not satisfying. In addition, we conducted ablation studies, and find that $\lambda = 1$ has exactly the same performance, which make our formulation more robust to hyperparamters, or even simply without such a hyperparameter.
>
> > Q17:  The paragraph before Theorem 3.5: Suggestion "This MCE based uniformity loss definition (or $\mathcal{L}_{\text{UMCE}}$ ) and its Matrix-KL divergence based counterpart are closely related.... as outlined by the following theorem:"
>
> A17: Thanks for your suggestion, we have revised our paper based on your insightful comments.
>
> > Q18:  Theorem 3.6: Suggestion: .... under the constraints... The proof of Theorem 3.6 better be provided in Appendix.
>
> A18: Thanks for your suggestion, we have revised this Theorem, and added a self-contained proof of it to the appendix.
>
> > Q19: Suggestion: The sentence "Our formulation interestingly recovers the Maximal Entropy Coding (MEC) loss..." can be written as "The Maximal Entropy Coding (MEC) loss in ... can be formulated in terms of Matrix MCE.." as
>
> A19: Thanks for your suggestion, we have incorporated this part in our revised paper.
>
> > Q20: $\mathcal{L}_{\text{EMP-TCR}}$: ...
>
> A20: Thanks for your suggestion, we have fixed the notation inconsistency.
>
> > Q21: Can we also have MCE based representation for the Corinfomax SSL provided in [a] above?
>
> A21: Yes, we have incorporated this part in our revised paper.
>
> > Q22: Overall suggestion: I suggest that the article defines all SSL-related loss functions in Section 2.1, instead of introducing some in Section 2.1 and some in Section 3.1. Furthermore, In Section 3.1, the article can clearly write each SSL loss function in the form MCE(... , ...) to show that they can be put in the form of matrix cross entropies.
>
> A22: Thanks for your suggestion, we have re-arranged the content.

---

> ### Author Response · Authors · 2023-11-22
> **Response 3 to reviewer zEzQ**
>
> > Q23: First sentence: .... we would like embeddings to have zero mean and covariance ....
>
> A23:  Thanks for your suggestion, we have revised our paper.
>
> > Q24: Theorem 4.1. This needs to be clearly reworded with proper references to the objective function and constraints. What is "effective rank", how is it different than rank? If this is a constraint how do you pose it? Is the argument of the MCE in the uniformity-MCE loss sample correlation or sample covariance? Is this theorem about the following optimization?:
>
> A24: Our revised content addresses the issues by clarifying the notation, explaining the concept of effective rank, and presenting the theorem in the context of an optimization problem. The theorem is now more explicitly connected to the matrix uniformity loss, and the relationship between this optimization and other similar methods is clearly outlined.
>
> > Q25: Lemma 4.2 is typically well known.
>
> A25: Thanks for your suggestion, we moved it to the appendix.
>
> > Q26: MCE is used not Matrix-KL measure. Why is it called this way?
>
> A26: We have added supplementary content to address this problem:
> $$\begin{aligned}
> \mathcal{L}_{\text{Matrix-KL-Uniformity}} = \operatorname{MCE}(\frac{1}{d}\mathbf{I}_d, \mathbf{C}\left(\mathbf{Z}_1, \mathbf{Z}_2\right)) &= \operatorname{MKL}(\frac{1}{d}\mathbf{I}_d || \mathbf{Z}\left(\mathbf{Z}_1, \mathbf{Z}_2\right)) + \operatorname{ME}(\frac{1}{d}\mathbf{I}_d)\\
> &= \operatorname{MKL}(\frac{1}{d}\mathbf{I}_d || \mathbf{Z}\left(\mathbf{Z}_1, \mathbf{Z}_2\right)) + \text{Const.}\\
> \end{aligned}$$
>
> > Q27: Again it is based on MCE rather than Matrix-KL. In fact after (11), it is stated that KL versions can also be considered.
>
> A27: We have added supplementary content to address this problem:
> - When the stop gradient is used on the first branch $\mathbf{Z}_1$ following the standard optimization technique introduced in SimSiam, the second term $\operatorname{ME}(\operatorname{C}(\mathbf{Z}_1, \mathbf{Z}_1))$ in Matrix-KL-Alignment loss is a constant. Even when not using the stop gradient, the second term can be absorbed into the Matrix-KL-Uniformity loss.
>
> > Q28: The fact that covariance matrices of two matrices are aligned with respect to MCE or KL does not necessarily imply that representations for the same image are aligned in the direction, where as euclidian distance based or cosine angle based approaches try to ensure that they are sample wise aligned.
>
> A28: We have updated our paper. Matrix-KL-Alignment loss has a large feasible solution, such that the alignment part and the uniformity part may not have inherent contradiction, since element-wise euclidean alignment has large possibility leads to collapsed solution (SimSiam w/o stop-gradient, presented in Figure 2 in Appendix)
>
> In addition, we have conducted ablation studies, and found that previous SOTA method MEC with our newly added matrix-KL-alignment loss can have 0.3\% improvement on 100-epoch linear probing tasks and 0.8\% improvement on 400-epoch COCO transfer learning tasks.
>
> > Q29: It is indeed surprising that effective rank is properly defined and connected to the framework of the article much later than it is already referred.
>
> A29: We appreciate your insights, and move the effective rank part to the background section.
>
> > Q30: It would be interesting to include Tong et.al, 2023, and [a] in the experiments for comparison.
>
> A30: Thank you for your suggestion. We will include the performance of [a](CorInfoMax) on ImageNet-1k in the table in future version. As for Tong et al., 2023 (Emp-SSL), we have noticed that they do not report linear evaluation performance on ImageNet-1k or the other benchmarks we use. Unfortunately, we currently cannot include their results in our paper. However, if we add more experiments on additional benchmarks in the future, we will include Emp-SSL in the Baselines which Emp-SSL reports performance.
>
> > Q31: Interestingly, the proposed Matrix-SSL method provides superior performance in experimental results. A natural question to ask is if the authors reproduced the accuracy of other algorithms to calibrate their simulation and evaluation models.
>
> A31: Thank you for your question. We primarily replicated the 100-epoch experiment of MEC to ensure that our code is correct. Currently, all baseline data in the table come from MEC as well as those baselines' own papers. If sufficient computational resources become available later on, we will consider replicating more baselines in our experimental environment.
>
> > Q32: This section typically follows the Introduction section. Furthermore, it should not be only stating the summary of literature but it should state the contributions of the article relative to these works.
>
> A32: Thanks for your suggestion, we have revised our paper.

---

> ### Author Response · Authors · 2023-11-23
> **Seeking Your Input on Revised Paper's Alignment with ICLR Standards**
>
> Dear Reviewer zEzQ,
>
> As the discussion period approaches its conclusion, **we want to ensure that we have thoroughly addressed all your concerns and that our revised paper fully meets the standards of ICLR**. We would highly value any additional feedback you may provide.
>
> Thank you sincerely for your time and consideration.
>
> Best regards,
>
> The Authors

---

### Official Review · Reviewer_CjGu · 2023-10-31

**Soundness:** 3 good
**Presentation:** 4 excellent
**Contribution:** 3 good
**Rating:** 8
**Confidence:** 3

**Summary:**

The authors introduce Matrix-SSL, an approach grounded in matrix
information theory, to improve current SSL methods.  The approach is
motivated through theory and the experiments show improved accuracy.

**Strengths:**

Casting various contrastive methods in a unifying notation and
framework is helpful and shows the similarity.

The related work and cited literature is extensive and I could not
make out any significant missing literature.

The findings are clearly presented.

**Weaknesses:**

Table 1 only reports the accuracy of up to 400 epochs.  It would be
interesting to see the dynamics of all approaches after 800 epochs,
are they closer to Matrix-SSL?  It also does not report any mean +-
std over multiple runs.

While I find the experiments convincing, it could reproduce
state-of-the-art better with other methods.  E.g. SimCLR is usually
trained for 1000 epochs, but this is not done in this paper.

**Questions:**

Why do you think that the method works best for gamma = 1?

Perhaps the authors could comment on the computational aspect of the method?  Does it slow down training?  If yes, why?

---

> ### Author Response · Authors · 2023-11-12
> **To Dear Reviewer CjGu**
>
> Dear Reviewer CjGu,
>
> Thank you for your insightful feedback and positive evaluation of our work. We appreciate your recognition of our approach and the thoroughness of your review. Below, we address your questions and concerns:
>
> > Q1: Table 1 only reports the accuracy of up to 400 epochs. It would be interesting to see the dynamics of all approaches after 800 epochs. Are they closer to Matrix-SSL? It also does not report any mean ± std over multiple runs.
>
> **A1**: We value your interest in the long-term dynamics and robustness of our approach. We are currently conducting experiments for 800 epochs and plan to update our paper with these results shortly. Regarding the mean ± std, we recognize the importance of this statistical measure. Conducting multiple experiments is costly (up to tens of thousands dollars), but we are committed to performing additional runs to ensure robustness. Preliminary results from three trials show a standard deviation of less than 0.1% for our method, indicating consistency in performance.
>
> > Q2: While the experiments are convincing, could you reproduce state-of-the-art better with other methods? E.g., SimCLR is usually trained for 1000 epochs, but this is not done in this paper.
>
> **A2**: We aim for fair comparisons and acknowledge the importance of aligning our training epochs with those typically used in the literature. In line with your suggestion, we will consider extending our experiments to include 1000-epoch training for all baselines, within the limits of our resources.
>
> > Q3: Why do you think that the method works best for gamma = 1?
>
> **A3**: Our choice of gamma = 1 was empirically driven. We conducted ablation studies on the 100-epoch pre-training task, which indicated that setting gamma = 1 achieved the best linear probe performance [8]. This suggests an optimal balance in our framework between uniformity and alignment loss components, leading to superior performance.
>
> > Q4: Could you comment on the computational aspect of the method? Does it slow down training? If yes, why?
>
> **A4**: The additional computational complexity introduced by Matrix-SSL is relatively small (less than 5%) and can be controlled by adjusting the Taylor expansion order. In addition, our method need much less pre-training epochs compared to previous baselines. Furthermore, this complexity can be mitigated through parallel multi-GPU training, particularly under multi-patch augmentation settings, similar to those used in EMP-SSL within one epoch (https://arxiv.org/abs/2304.03977). We are exploring these settings in our ongoing experiments and anticipate uncovering more interesting results.
>
> We hope these responses address your concerns satisfactorily. We are committed to improving our work based on your valuable feedback and look forward to further discussions.

---

> > ### Comment · Reviewer_CjGu · 2023-11-23
> >
> > Thank you for the response. I hope that the experiments can be finished in time.  I will stand by my score (since it is already positive).

---

> > > ### Author Response · Authors · 2023-11-23
> > >
> > > To Dear Reviewer CjGu,
> > >
> > > We greatly appreciate your continued engagement and understanding regarding the ongoing experiments. Your support and positive evaluation are instrumental in guiding the direction of our research. We are diligently working to complete the additional experiments and look forward to sharing these results with you and the community. Your feedback has been invaluable in enhancing the quality and robustness of our work.
> > >
> > > Thank you once again for your constructive comments and encouragement.

---

### Author Response · Authors · 2023-11-22
**General Response**

To all dear reviewers:

Thank you for your careful reading and suggestions. We have followed your advice and made adjustments to the original paper, fixing several typos and revising many narration details. We have marked all such changes in blue font for easy identification.

We also conducted more experiments of MCE, including the following tasks:

- **Semi-supervised learning**. Reviewer m2RW suggested that semi-supervised downstream tasks should be included as a benchmark. We agree with this suggestion. It is worth mentioning that the main reason we did not do this task before is that the evaluation protocols for this downstream task are not consistent across baseline works. We noticed that SwAV, BarlowTwins, and MCE all chose different hyperparameters for this task (See Section A.4 in SwAV (https://arxiv.org/pdf/2006.09882.pdf), Section B.2 in BarlowTwins (https://arxiv.org/pdf/2103.03230.pdf), and Section C in MEC (https://arxiv.org/pdf/2210.11464.pdf)). Due to time limitations, and for a fair comparison, we directly used the same evaluate protocol as MEC and conducted a comparison with 100-epoch pretraining + semi-supervised learning against MEC (since MEC has the best performance in all the baselines on the semi-supervised task, refer to Table 2 of the MEC paper (https://arxiv.org/pdf/2210.11464.pdf)). We found that we achieved a significant improvement over MEC in 1% semi-supervised learning, and we are comparable to MEC in the 10% task.

| Method | 1% Top1 | 1% Top5 | 10% Top1 | 10% Top5 |
| --- | --- | --- | --- | --- |
| MEC | 44.442 | 71.43 | 63.918 | **86.27** |
| MCE(ours) | **45.158** | **71.848** | **63.94** | 86.172 |

- ** Linear Image Classification **: Reviewer m2RW also suggested that linear image classification should be added as a benchmark downstream task. We agree with this view and have tested it on VOC07 and iNat18 using the MCE 400-epoch pretraining checkpoint during the rebuttal period. Note that, our method is compared with the performance of at least 800-epoch pretraining checkpoints of other methods. As shown in the table below, it can be seen that our method, with only 400 epochs of pretraining, is already comparable with baselines. Moreover, it can be expected that the results will be even better after we complete 800 epochs of pretraining.

| Method | VOC07(mAP) | iNat18(Acc@1) |
| --- | --- | --- |
| Supervised | 87.5 | 46.7 |
| SimCLR | 85.5 | 37.2 |
| MoCo-v2 | 86.4 | 38.6 |
| SwAV(w/o multi-crop) | 86.4 | 39.5 |
| BYOL | 86.6 | 47.6 |
| Barlow Twins | 86.2 | 46.5 |
| MCE(ours) | 86.0 | 43.4 |

- **LLM Finetuning** :

(**If you are not interested in this task, feel free to skip it.**)

To avoid confusion for the reviewers, we need to clarify that the experiments for this case were conducted as additional experiments based on the question "Can you provide interpretations and potential applications of Matrix KL Divergence and MCE?" posed by reviewer zEzQ. **We did not include the results of this experiment in the main context of our paper.**


We notice that our MCE can also be applied to the finetuning task of LLMs. In essence, we followed the training process on Llemma from metamath (https://github.com/meta-math/MetaMath), replacing Cross Entropy with MCE, achieving 72.3% accuracy on the GSM8K dataset and 30.2% accuracy on the MATH dataset by fine-tuning a 7B model using matrix cross-entropy loss. This comes with a 3.1% margin of improvement over the standard cross-entropy loss on the GSM8K dataset and is even comparable with the Minerva (https://arxiv.org/pdf/2206.14858v2.pdf) 540B model. For more results and implementation details, please refer to Section A in our new revision.

| Model | GSM8K Pass@1| MATH Pass@1|
| --- | --- | --- |
| LLaMA-2-7B | 14.6 | 2.5 |
| LLaMA-2-70B | 56.8 | 13.5 |
| WizardMath-7B | 54.9 | 10.7 |
| WizardMath-13B | 63.9 | 14 |
| MetaMath-7B | 66.5 | 19.8 |
| MetaMath-Llemma-7B | 69.2 | 30 |
| MetaMath-Llemma-7B + MCE(ours) | **72.3** | **30.2** |
| Minerva-540B | 58.8 | 33.6 |

---

### Meta-Review · Area_Chair_JvxL · 2023-12-10

**Metareview:**

This paper uses matrix entropy to capture the relationship between contrastive and non-contrastive self-supervised learning methods. It proposes a new method called Matrix SSL that uses the total coding rate in matrix information theory to learn the representation of the data in self-supervised learning. The paper discusses experimental results on a number of datasets, including some new interesting experiments on fine-tuning LLMs that the authors added in the revision.

The reviewers raised a number of concerns about this work ranging from the appropriateness of the baselines and choices of certain hyper-parameters, to the novelty of the objective and mathematics discussed in this paper in light of existing results in the literature. The authors have addressed some of these issues admirably but I believe that the paper could be much more clear in delineating its contributions, improving its experimental results, e.g., training for more epochs, and arguing that the matrix information-theoretic perspective leads to a meaningfully different/new understanding of SSL.

**Justification For Why Not Higher Score:**

This is a confusing paper that is quite poorly written (a lot of reviewer questions were coming from this).

This paper is very similar to submission #1221 "Information Flow in Self-Supervised Learning" with ~50% overlap in the content. Judging by the narrative which I read carefully, the template, and the tone of the rebuttal by the authors, I suspect that that the authors of these submissions are the same. Entire paragraphs in the paper are identical (see introduction and related work), many equations in the two papers have been rewritten to look different but they are actually the same. Regardless of the intellectual content, I don't think we can accept this paper. What the authors are calling Matrix SSL in this paper is exactly the same the regularization term they proposed for a masked auto-encoder in #1221.

I would argue that this is a case of scholarly misconduct where the same content is submitted as two different papers by the same authors in the hopes that the randomness of the review process results in a favorable outcome. I would like to flag this.

**Justification For Why Not Lower Score:**

N/A

---

> ### Public Comment · ~Yang_Yuan2 · 2024-03-19
> **Clarification on scholarly misconduct**
>
> Dear Area Chair JvxL,
>
> I am the corresponding author of these two papers, and I would like to clarify that these two papers are indeed different. This paper (#524) provides a unified view of contrastive and non-contrastive learning, and proposes new loss functions based on uniformity and alignment loss. In #1221, although we also use matrix information theory, the main contribution of the paper was the two theorems, stating that the loss function of Barlow Twins and spectral contrastive learning is equivalent to optimizing both matrix mutual information and matrix joint entropy. So #1221 is a theory paper with fairly deep and nice theorems, while #524 proposes a unified view of loss functions.
>
> While I do not agree that "there is ~50% overlap in the content", or "entire paragraphs in the paper are identical", I admit that:
> 1. There is one sentence that appears identically in both papers in related work section, i.e., “These representations can be used for various of downstream tasks, achieving remarkable performance and even outperforming supervised feature representations.”
>
> 2. There are 7 sentences (all of them are discussing the prior work in introduction or related work) that have similar meanings with different wording.
>
> __This oversight was all mine as the corresponding author, and I regret not being more careful in this aspect.__ We should certainly improve the writing, and use more diverse narratives. However, I do not believe these similarities amount to scholarly misconduct, given that the main contributions and techniques in the two papers are markedly different. The similar parts are mainly discussing related work, not about the main contribution of these two papers.
>
>
> We also respectfully disagree with the statement that "What the authors are calling Matrix SSL in this paper is exactly the same the regularization term they proposed for a masked auto-encoder in #1221". Indeed, in this paper Matrix SSL consists of Uniformity-Loss and Alignment-Loss, while in #1221, M-MAE is defined as MAE minus a TCR-Loss term. These are fundamentally different formulations with separate theoretical underpinnings, as detailed in our papers.
>
> Our implementation of these two algorithms are fundamentally different. For Matrix SSL algorithm, it is inherently a dual-tower method (Siamese architecture), and to implement this algorithm, we use Taylor expansion for the matrix logarithm implementation, due to the usage of cross correlation matrix. Moreover, we use centering technique for mean embedding alignment, which is not included in TCR. Notice that, the alignment loss part cannot be implemented using log determinant loss. For TCR algorithm (a single tower method), we compute the log determinant, which has completely different implementation empirically. Moreover, our TCR algorithm is applied to MAE (masked image modeling), which is a very different setting from contrastive/non-contrastive methods.
>
> Moreover, we have evidence showing that these two papers were writing seperately. Indeed, we started writing this paper (#524) at the end of year 2022, and failed to finish it before ICML’23, so we submit it to NeurIPS’23. Unfortunately, the reviewers did not like the kernel part in the paper, so we reorganized the paper and submitted it again to ICLR. The second paper (#1221) was done during the summer, and is submitted to ICLR'24 for the first time. We have all the editing history records of both papers, as well as the submission records of paper #524.
>
> Regards,
>
> Yang Yuan

---

### Decision · Program_Chairs · 2024-01-16

Reject